**Title page**
**Modeling inorganic carbon dynamics in the Seine River**
**continuum in France**
Audrey Marescaux[1], Vincent Thieu[1], Nathalie Gypens[2], Marie Silvestre[3], Josette Garnier[1]
[1]Sorbonne Université, CNRS, EPHE, Institut Pierre Simon Laplace FR 636, UMR 7619 METIS,
Paris, France
[2]Université Libre de Bruxelles, Ecologie des Systèmes Aquatiques, Brussels, Belgium
[3]Sorbonne Université, CNRS, Federation Ile-de-France of Research for the Environment FR3020,
Paris, France
Correspondence email : Audreymarescaux@gmail.com
Article revised to be submitted to Hydrology and Earth System Sciences

## Abstract

Inland waters are an active component of the carbon cycle where transformations and transports are associated with carbon dioxide ($CO_2$) outgassing. This study estimated $CO_2$ emissions from the human-impacted Seine River (France) and provided a detailed budget of aquatic carbon transfers for organic and inorganic forms, including in-stream metabolism along the whole Seine River network. The existing process-based biogeochemical pyNuts-Riverstrahler model was supplemented with a newly developed inorganic carbon module and simulations were performed for the recent time period 2010-2013. New input constraints for the modelling of riverine inorganic carbon were documented by field measurements and complemented by analysis of existing databases. The resulting dissolved inorganic carbon (DIC) concentrations in the Seine aquifers ranged from 25 to 92 mgC $L^{-1}$, while in wastewater treatment plant (WWTP) effluents our DIC measurements averaged 70 mgC $L^{-1}$.

Along the main stem of the Seine River, simulations of DIC, total alkalinity, pH, and $CO_2$ concentrations were of the same order of magnitude as the observations, but seasonal variability was not always well reproduced. Our simulations demonstrated the $CO_2$ supersaturation with respect to atmospheric concentrations over the entire Seine River network. The most significant outgassing was in lower order streams while peaks were simulated downstream of the major WWTP effluent. For the period studied (2010–2013), the annual average of simulated $CO_2$ emissions from the Seine drainage network were estimated at $364 \pm 99$ Gg C $yr^{-1}$.

Results from metabolism analysis in the Seine hydrographic network highlighted the importance of benthic activities in headwaters while planktonic activities occurred mainly downstream in larger rivers. The net ecosystem productivity remained negative throughout the 4 simulated years and over the entire drainage network, highlighting the heterotrophy of the basin.

**Keywords:** $CO_2$ outgassing; inorganic carbon modeling; instream metabolisms; waste- and ground water inputs; carbon budget ; temperate Seine River

**Graphical abstract:**

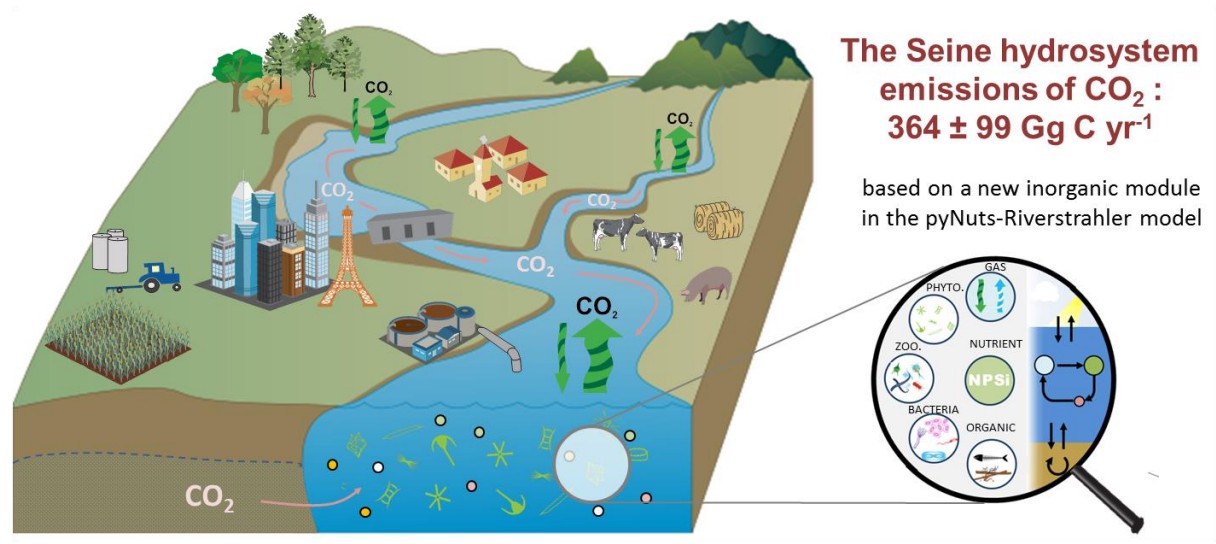

**Highlights:**

- $CO_2$ emission from the Seine River was estimated at $364 \pm 99$ GgC yr$^{-1}$ with the Riverstrahler model.
- $CO_2$ riverine concentrations are modulated by groundwater discharge and instream metabolism.
- $CO_2$ emissions account for 31% of inorganic carbon exports, the rest being exported as DIC.

# 1. Introduction

Rivers have been demonstrated to be active pipes for transport, transformation, storage and outgassing of inorganic and organic carbon (Cole et al., 2007). Although there are large uncertainties in the quantification of flux from inland waters, carbon dioxide ($CO_2$) outgassing has been estimated to be a significant efflux to the atmosphere, subjected to regional variabilities (Cole et al., 2007; Battin et al., 2009a; Aufdenkampe et al., 2011; Lauerwald et al., 2015; Regnier et al., 2013; Raymond et al., 2013a; Sawakuchi et al., 2017; Drake et al., 2017). These variabilities are determined by regional climate and watershed characteristics and are related to terrestrial carbon exports under different forms, from organic to inorganic, and dissolved to particulate. Organic carbon entering rivers can originate from terrestrial ecosystems as plant detritus, soil leaching or soil erosion and groundwater supply, but it can also be produced instream by photosynthesis or brought by dust particles (Prairie and Cole, 2009; Drake et al., 2017). Inorganic carbon originate from groundwater, soil leaching and exchange by diffusion at the air–water interface, depending on the partial pressure of $CO_2$ ($pCO_2$) at the water surface with respect to atmospheric $pCO_2$ (Cole et al., 2007; Drake et al., 2017; Marx et al., 2018). Beside air-water exchanges, carbon exchanges occur at the water–sediment interface, through biomineralization and/or burial (Regnier et al., 2013b). As a whole, eutrophic, oligo- and mesotrophic hydrosystems generally act as a source of carbon however, lentic systems may be undersaturated with respect to atmospheric $pCO_2$ (Prairie and Cole, 2009; Xu et al., 2019; Yang et al., 2019).

Direct measurements of $pCO_2$ or isotopic surveys (as realized by Dubois et al. 2010 in the Mississippi River) along the drainage network are still too scarce to accurately support temporal and spatial analyses of $CO_2$ variability. While calculations from pH, temperature and alkalinity may help reconstruct spatiotemporal patterns of $CO_2$ dynamics (Marescaux et al., 2018b), modeling tools can predict the fate of carbon in whole aquatic systems. Indeed, modeling

approaches have made it possible to simulate and quantify carbon fluxes between different
reservoirs: atmosphere, biosphere, hydrosphere and lithosphere (e.g., Bern-SAR, Joos et al.,
1996; ACC2, Tanaka et al., 2007; TOTEM, Mackenzie et al., 2011; MAGICC6, Meehl et al.,
2007). In addition to these box approaches, a number of more comprehensive mechanistic
models, describing biogeochemical processes involved in carbon cycling and $CO_2$ evasion,
have been set up for oceans (e.g., Doney et al., 2004; Aumont et al., 2015), coastal waters (e.g.,
Borges et al., 2006; Gypens et al., 2004, 2009, 2011) and estuaries (e.g., Cai and Wang, 1998;
Volta et al., 2014, Laruelle et al., 2019). In inland waters, the NICE-BGC model (Nakayama,
2016) accurately represents $CO_2$ evasion at the global scale. However, to our knowledge, while
several process-based river models  describe the carbon cycle through organic matter input and
degradation by aquatic microorganisms (e.g., PEGASE, Smitz et al., 1997; ProSe, Vilmin et
al., 2018; QUAL2Kw, Pelletier et al., 2006; QUAL-NET, Minaudo et al., 2018, QUASAR,
Whitehead et al., 1997; Riverstrahler, Billen et al., 1994; Garnier et al., 2002), none of them
describes the inorganic carbon cycle including carbon dioxide outgassing.
The Seine River (northwestern France) has long been studied using the biogeochemical riverine
Riverstrahler model (Billen et al., 1994; Garnier et al., 1995), a generic model of water quality
and biogeochemical functioning of large river systems. For example, the model has made it
possible to quantify deliveries to the coastal zone and understand eutrophication phenomena
(Billen and Garnier, 2000; Billen et al., 2001; Passy et al., 2016; Garnier et al., 2019), nitrogen
transformation and $N_2O$ emissions (Garnier et al., 2007, 2009; Vilain et al., 2012) as well as
nitrate retention (Billen and Garnier, 2000; Billen et al., 2018), and organic carbon metabolism
(Garnier and Billen, 2007; Vilmin et al., 2016). It is only recently that we investigated $pCO_2$
and emphasized the factors controlling $pCO_2$ dynamics in the Seine River (Marescaux et al.,
2018b) or its estuary (Laruelle et al., 2019).
The purpose of the present study was to quantify the sources, transformations, sinks and gaseous
emissions of inorganic carbon using the Riverstrahler modelling approach (Billen et al., 1994;
Garnier et al., 2002; Thieu et al., 2009). A further aim in newly implementing this $CO_2$ module
was to quantify and discuss autotrophy versus heterotrophy patterns in regard to $CO_2$
concentrations and supersaturation in the drainage network.

## 2. Material and methods

### 2.1. Description of the Seine basin

Situated in northwestern France, 46°57' – 50°55' north and 0°7' 1" – 4° east, the Seine basin
(~76,285 km²) has a temperate climate and a pluvio-oceanic hydrologic regime (Figure 1). The
mean altitude of the basin is 150 m above sea level (ASL) with 1% of the basin reaching more
than 550 m ASL in the Morvan (Guerrini et al., 1998). The water flow at Poses (stream order
7, basin area 64,867 km²), the most downstream monitoring station free from tidal influence,
averaged 490 m$^3$ s$^{-1}$ during the 2010–2013 period (the HYDRO database,
http://www.hydro.eaufrance.fr, last accessed 2020/02/11). The major tributaries include the
Marne and upper Seine rivers upstream from Paris, and the Oise River downstream from Paris
(Figure 1a). Three main reservoirs, storing water during winter and sustaining low flow during
summer, are located upstream on the Marne River and the upstream Seine and its Aube tributary
(Figure 1a). The total storage capacity of these reservoirs is 800 10$^6$ m$^3$ (Garnier et al., 1999).
The maximum water discharge of these tributaries occurs during winter with the lowest
temperature and rate of evapotranspiration; the opposite behavior is observed during summer
(Guerrini et al., 1998).
Except for the crystalline rocks in the north and from the highland of the Morvan (south), the
Seine basin is for the most part located in the lowland Parisian basin with sedimentary rocks
(Mégnien, 1980; Pomerol and Feugueur, 1986; Guerrini et al., 1998). The largest aquifers are

in carbonate rock (mainly limestone and chalk) or detrital (sand and sandstone) material separated by impermeable or less permeable layers.

The concept of Strahler stream order (SO) (Strahler, 1957) was adopted for describing the geomorphology of a drainage network in the Riverstrahler model (Billen et al., 1994). The smaller perennial streams are order 1. Only confluences between two river stretches with the same SO produce an increase in Strahler ordination (SO+1) (Figure 1a). The mean hydrophysical characteristics of the Seine River are aggregated by stream orders shown in Table 1.

The Seine basin is characterized by intensive agriculture (more than 50% of the basin, CLC - EEA, 2012) is mostly concentrated in the Paris conurbation (12.4 million inhabitants in 2015) (Figure 1) (INSEE, 2015). Located 70 km downstream of Paris, the largest wastewater treatment plant in Europe (Seine Aval, SAV WWTP) can treat up to $6 \cdot 10^6$ inhab eq per day, releasing 15.4 $m^3$ $s^{-1}$ into the lower Seine River (*Syndicat interdépartemental pour l'assainissement de l'agglomération parisienne*; French acronym SIAAP, http://www.siaap.fr/, last accessed 2020/02/11).

| SO | Draining area $km^2$ | Cum. length $Km$ | Width (*) $m$ | Depth (**) $m$ | Slope (*) $m\ m^{-1}$ | Discharge (**) $m3\ s^{-1}$ | Flow velocity (**) $m\ s^{-1}$ |
|----|----------------------|------------------|---------------|----------------|-----------------------|------------------------------|--------------------------------|
| 1  | 36083                | 12759            | 2.4           | 0.14           | 0.01442               | 0.13                         | 0.34                           |
| 2  | 12354                | 5231             | 5.2           | 0.29           | 0.00540               | 0.66                         | 0.36                           |
| 3  | 7067                 | 2871             | 10.6          | 0.45           | 0.00300               | 2.17                         | 0.47                           |

| | | | | | | | |
|---|---|---|---|---|---|---|---|
| 4 | 4054 | 1548 | 20.2 | 0.79 | 0.00212 | 6.35 | 0.33 |
| 5 | 2649 | 943 | 46.0 | 1.11 | 0.00060 | 25.87 | 0.46 |
| 6 | 2094 | 636 | 77.8 | 2.51 | 0.00029 | 82.22 | 0.42 |
| 7 | 1354 | 318 | 168.3 | 2.61 | 0.00037 | 416.16 | 0.81 |

*Table 1: Hydro-morphological characteristics of the Seine drainage network, (\*) averaged by Strahler order*

*(SO) and (\*\*) over the time period 2010-2013. Hydrographic network provided by the Agence de l'Eau Seine*

*Normandie and water discharges by the national Banque Hydro database. Depth and flow velocity calculated*

*according to Billen et al 1994; width calculated according to Thieu et al., 2009.*

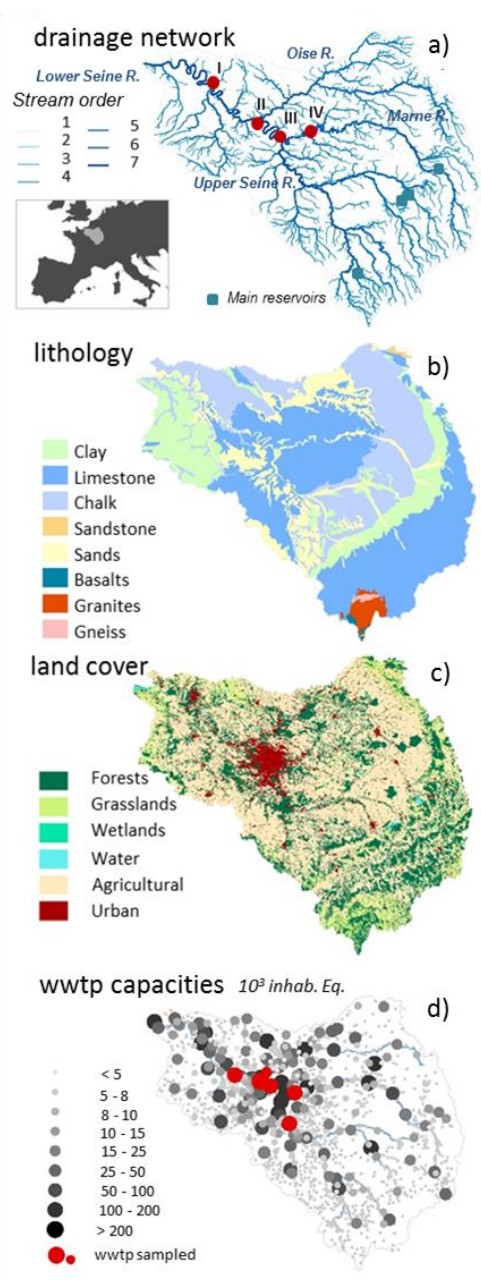


*Figure 1 Characteristics of the Seine basin: a) drainage network according to Strahler*

*stream orders (Strahler, 1952, 1957), monitoring stations (I: Poses, II: Poissy (Downstream*

*Paris), III: Paris, IV: Ferté-sous-Jouarre (Upstream Paris); b) the lithology according to*

*Albinet, (1967); c) land use according to the Corine Land Cover database, with six simplified*

*classes (EEA, 2012); d) wastewater treatment plants (WWTPs) of the basin. Red dots are the*

*WWTPs sampled in 2018.*

## 2.2. The pyNuts-Riverstrahler model and its biogeochemical model, RIVE

*The biogeochemical model, RIVE*. The core of the biogeochemical calculation of the pyNuts-
Riverstrahler model (described hereafter) is the RIVE model (e.g., Billen et al., 1994; Garnier
et al., 1995; Garnier et al., 2002; Servais et al., 2007) (https://www.fire.upmc.fr/rive/), which
simulates concentrations of oxygen, nutrients (nitrogen (N), phosphorus (P) and silica (Si)),
particulate suspended matter, and dissolved and particulate organic carbon (three classes of
biodegradability) in a homogeneous water column. Biological compartments are represented
by three taxonomic classes of phytoplankton (diatoms, Chlorophyceae and Cyanobacteria), two
types of zooplankton (rotifers with a short generation time and microcrustaceans with a long
generation time), two types of heterotrophic bacteria (small autochthonous and large
allochthonous with a higher growth rate than the small ones), as well as two types of nitrifying
bacteria (ammonium-oxidizing bacteria and nitrite-oxidizing bacteria).
The model also describes benthic processes (erosion, organic matter degradation,
denitrification, etc.) and exchanges with the water column with the explicit description of
benthic organic matter, inorganic particulate P and benthic biogenic Si state variables. The
benthic component does not explicitly represent all the anaerobic reduction chains,
denitrification being the major anaerobic microbial process.
A detailed list of the state variables of the RIVE model is provided in S1. Most of the kinetic
parameters involved in this description have been previously determined through field or
laboratory experiments under controlled conditions and are fixed a priori (see detailed
description of all kinetics and parameters values in Garnier et al., 2002). To date, there has been
no explicit representation of inorganic carbon in the RIVE model (see this new input in S1).
Riverstrahler allows for the calculation of water quality variables at any point in the aquatic
continuum based on a number of constraints characterizing the watershed, namely, the
geomorphology and hydrology of the river system and the point and diffuse sources of nutrients.
*Geomorphology*. A drainage network can be described as subbasins (tributaries) connected to
one or several main axes, that define a number of modelling units. The modelling approach
considers the drainage network as a set of river axes with a spatial resolution of 1 km (axis-
object), or they can be aggregated to form subbasins that are idealized as a regular scheme of
tributary confluences where each stream order is described by mean characteristics (basin-
object). Here, the Seine drainage network starts from headwater until it fluvial outlet (Poses)
and was divided into 69 modeling units, including six axes (axis-object) and 63 upstream basins
(basin-object). A map and a table introducing the main characteristics of the modeling units are
provided in S2.
*Hydrology*. Runoffs were calculated over the whole Seine basin using water discharge
measurements at 48 gauged stations (source: Banque Hydro database,
http://www.hydro.eaufrance.fr/, last accessed 2020/02/11). Surface and base flow contributions
were estimated applying the BFLOW automatic hydrograph separation method (Arnold and
Allen, 1999) over the recent time series of water discharges (2010–2017). For the study period
(2010–2013), the mean base flow index (BFI = 0.71) of the Seine basin indicates the extent of
the groundwater contribution to river discharge, with spatial heterogeneity following the main
lithological structures (Figure 1b), but when summarizing the BFI criteria by Strahler order,
significant differences did not appear (not shown).
*Water temperature.* Water temperature was calculated according to an empirical relationship,
adjusted on inter-annual averaged observations (2006—2016), and describes seasonal variation
of water temperature in each Strahler order with a 10-day time step (see S2).
*Diffuse and point sources.* Riverstrahler manages the calculation of the RIVE model according
to a Lagrangian routing of water masses along the hydrographic network (Billen et al., 1994)
and is a generic model of water quality and biogeochemical functioning of large drainage
networks that simulates water quality. PyNuts is a modeling environment that can calculate the
constraints (diffuse and point sources) on the Riverstrahler model at a multiregional scale
(Desmit et al., 2018 for the Atlantic façade).

## 2.2.1.    Development of an inorganic carbon module

**Introducing the carbonate system**

The carbonate system was described by a set of equations  (named $CO_2$-module) based on a
previous representation provided by Gypens et al. (2004) and adapted for freshwater
environments (N. Gypens and A.V. Borges, personal communication). This $CO_2$-module was
fully integrated in the RIVE model (Figure 2). It aims at computing the speciation of the
carbonate system based on two new state variables: dissolved inorganic carbon (DIC) and total
alkalinity (TA), making it possible to calculate carbon dioxide ($CO_2$). The module uses three
equations (see S3: Eqs. 1, 2, 3) that also calculate bicarbonate ($HCO_3^-$), carbonate ($CO_3^{2-}$) and
hydronium ($H_3O^+$). Indeed, two variables of the carbonate system are sufficient to calculate all
the other components (Zeebe and Wolf-Gladrow, 2001). Here, DIC and TA were selected
because the biological processes involved in their spatiotemporal variability along the aquatic
continuum were already included in the RIVE model (Figure 2). We calculated pH as a function
of TA and DIC using the Culberson equation (Culberson, 1980) (S3.4).

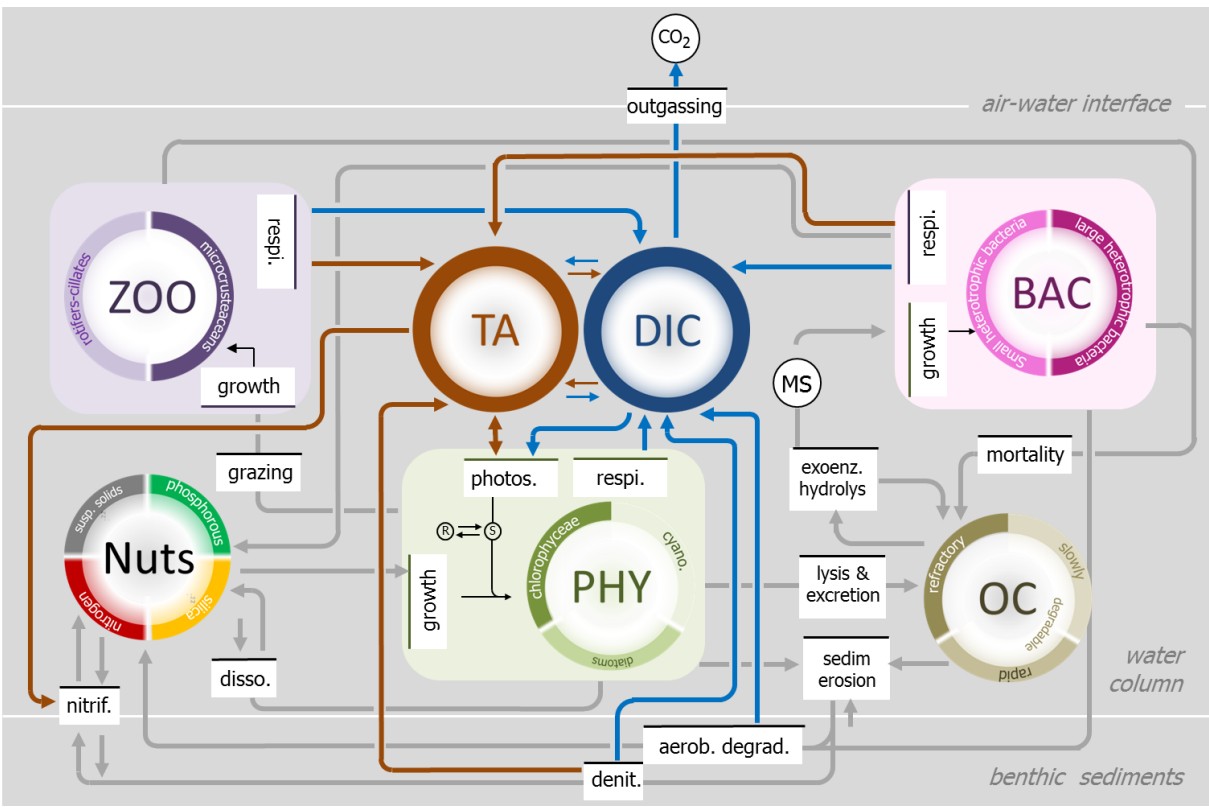

*Figure 2 Schematic representation of the ecological RIVE model (inspired from Billen et al.*
*1994, Garnier & Billen, 1994), with grey lines indicating the main processes simulated in the*
*water column and at the interface with sediment (oxygen not shown), and implementation of*
*the new inorganic module, based on total alkalinity (TA, maroon) and dissolved inorganic*
*carbon (DIC, blue).*

## Aquatic processes affecting TA and DIC

The exchange of $CO_2$ between the water surface and the atmosphere depends, respectively, on the gas transfer velocity (k-value) and on the sign of the $CO_2$ concentration gradient at the water surface–atmosphere interface (S3.5). Change in $pCO_2$ will in turn affect DIC concentrations (see Table 2, Eq. 1). Dissolved or particulate organic matter is mostly degraded by microbial activities (more or less quickly depending on their biodegradability), resulting in $CO_2$ and $HCO_3^-$ production (Servais et al., 1995), thus inducing a change in DIC and TA concentrations in the water column (Table 2, Eq. 2, Figure 2). Photosynthesis and denitrification processes also affect DIC and TA (Table 2, Eqs. 3–5), while instream nitrification only influences TA (Table 2, Eq. 6, Figure 2).

*Table 1 Stoichiometry of the biogeochemical processes, influencing dissolved inorganic carbon (DIC) and total alkalinity (TA) in freshwater, as taken into account in the new inorganic carbon module. TA and DIC expressed in mol:mol of the main substrate (either C or N).*

| Process | Equation | DIC | TA | Eq. |
|---------|----------|-----|-----|-----|
| FCO$_2$ | $CO_2(aq) \leftrightarrow CO_2(g)$ | ±1 | 0 | 1 |
| Aerobic degradation | $C_{106}H_{263}O_{11}N_{16}P + 106O_2$ $\rightarrow 92CO_2 + 14HCO_3^- + 16NH_4^+ + HPO_4^{2-} + 92H_2O$ | +1 | +14/106 | 2 |
| Photosynthesis (NO$_3^-$ uptake) | $106CO_2 + 16NO_3^- + H_2PO_4^- + 122H_2O + 17H^+$ $\rightarrow C_{106}H_{263}O_{11}N_{16}P + 138O_2$ | -1 | +17/106 | 3 |
| Photosynthesis (NH$_4^+$ uptake) | $106CO_2 + 16NH_4^+ + H_2PO_4^- + 106H_2O$ $\rightarrow C_{106}H_{263}O_{11}N_{16}P + 106O_2 + 15H^+$ | -1 | -15/106 | 4 |
| Denitrification | $5CH_2O + 4NO_3^- + 4H^+ \rightarrow 5CO_2 + 2N_2 + 7H_2O$ | +1 | +4/5 | 5 |
| Nitrification | $NH_4^+ + 2O_2 \rightarrow 2H^+ + H_2O + NO_3^-$ | 0 | -2 | 6 |

**State equations and parameters of the inorganic carbon module**

These processes affecting TA and DIC result in equations governing inorganic carbon dynamics as:

$$TA = TA_{t-1} + dt.\frac{dTA}{dt} + TA_{inputs} \qquad \text{Eq. 7}$$

with:

$$\frac{dTA}{dt} = \left(\frac{14}{106}\frac{(respbact + respZoo + respBent)}{M(C)}\right.$$

$$+ \left(\frac{4}{5}Denit - 2.nitr['AOB']\right).M(N)^{-1} + \left(\frac{17}{106}\frac{uptPhyNO_3^-}{uptPhyN}\right. \qquad \text{Eq. 8}$$

$$\left.\left. - \frac{15}{106}\frac{uptPhyNH_4^+}{uptPhyN}\right).phot.M(O_2)^{-1}\right)1000$$

where $TA_{t-1}$ is the value of TA ($\mu$mol L$^{-1}$) in the previous time step (t−1). $Respbact$, $RespZoo$, and $respBent$ are respectively the heterotrophic planktonic respiration of bacteria, zooplankton and benthic bacteria already included in RIVE (mgC L$^{-1}$ h$^{-1}$). $M(C)$ is the molar mass of the carbon (12 g mol$^{-1}$). $Denit$ and $nitr['AOB']$ are respectively the processes of denitrification and nitrification by ammonia-oxidizing bacteria (AOB) as implemented in the RIVE model (mgN L$^{-1}$ h$^{-1}$); $M(N)$ is the molar mass of the nitrogen (14 g mol$^{-1}$). $phot$ is the net photosynthesis (mgO$_2$ L$^{-1}$ h$^{-1}$). $uptPhyN$ is the nitrogen uptake by phytoplankton (mgN L$^{-1}$ h$^{-1}$) which is differentiated for nitrate ($uptPhyNO3^-$, mgC L$^{-1}$ h$^{-1}$) and ammonium ($uptPhyNH4^+$, mgC L$^{-1}$ h$^{-1}$), and $M(O_2)$ is the molar mass of the dioxygen (32 g mol$^{-1}$). TA$_{inputs}$ is TA ($\mu$mol L$^{-1}$) entering the water column by diffuse sources (groundwater and subsurface discharges) and point sources (WWTPs).

$$DIC = DIC_{t-1} + dt.\frac{dDIC}{dt} + \text{DIC}_{\text{inputs}} \qquad \text{Eq. 9}$$

with:

$$\frac{dDIC}{dt} = (respbact + respZoo + respBent) + denit.M(C).M(N)^{-1}$$

$$\text{Eq. 10}$$

$$+ phot.M(C).M(O_2)^{-1} + \frac{F_{CO_2}}{depth}$$

where $DIC_{t-1}$ is the value of DIC (mgC L$^{-1}$) in the previous time step (t−1). $F_{CO_2}$ is the CO$_2$
flux at the water–atmosphere interface in mgC m$^{-2}$ h$^{-1}$ described in S3.5; depth is the water
column depth (m).
The different values of constants and parameters used in the inorganic carbon module are
introduced in Table 1 of S3.6. The full inorganic carbon module is described in S3 (3.1 to 3.6).

## 2.2.2.    Input constraints of the pyNuts-Riverstrahler model

**Diffuse sources from soil and groundwater**

Diffuse sources are calculated at the scale of each modeling units, based on several spatially
explicit databases describing natural and anthropogenic constraints on the Seine River basin.
Diffuse sources are taken into account by assigning a yearly mean concentration of carbon and
nutrients to subsurface and groundwater flow components, respectively. These concentrations
are then combined with a 10-day time step description of surface and base flows to simulate the
seasonal contribution of diffuse emissions to the river system. For nutrients, several applications
of the Riverstrahler on the Seine River basin refined the quantification of diffuse sources: e.g.,
Billen and Garnier (2000) and Billen et al. (2018) for nitrogen; Aissa-Grouz et al. (2016) for
phosphorus; Billen et al. (2007), Sferratore et al. (2008) and Thieu et al. (2009) for N, P and Si.
In this study we revised our estimates for diffuse organic carbon sources and propose TA and
DIC values for the Seine basin. The summary of all the carbon-related inputs of the model is
provided in Table 3.
Dissolved organic carbon (DOC) input concentrations were extracted from the AESN database
(http://www.eau-seine-normandie.fr/, last accessed 2020/02/11) and averaged by land use for
subsurface sources (mean, 3.13 mgC $L^{-1}$; sd, 4.56 mgC $L^{-1}$; 3225 data for 2010–2013). For
groundwater sources, concentrations were extracted from the ADES database
(www.ades.eaufrance.fr, last accessed 2020/02/11) and averaged by MESO waterbodies
(French acronym: Masse d'Eau SOuterraine, see S4; mean, 0.91 mgC $L^{-1}$; sd, 0.8 mgC $L^{-1}$;
16,000 data for 2010–2013). These concentrations were separated into three pools of different
biodegradability levels, with 7.5% rapidly, 17.5% slowly biodegradable and 75% refractory
DOC for subsurface sources and 100% refractory DOC for groundwater flow (Garnier,
unpublished).
Total POC inputs were calculated based on estimated total suspended solid (TSS) fluxes,
associated with a soil organic carbon (SOC) content provided by the LUCAS Project (samples
from agricultural soil, Tóth et al., 2013), the BioSoil Project (samples from European forest
soil, Lacarce et al., 2009) and the Soil Transformations in European Catchments (SoilTrEC)
Project (samples from local soil data from five different critical zone observatories (CZOs) in
Europe, Menon et al., 2014) (Aksoy et al., 2016). TSS concentrations were calculated using
fluxes of TSS provided by WaTEM-SEDEM (Borrelli et al., 2018) and runoffs averaged over
the 1970–2000 period (SAFRAN-ISBA-MODCOU, SIM; Habets et al., 2008). The POC mean
was 8.2 mgC $L^{-1}$; sd, 10.4 mgC $L^{-1}$ in subsurface runoff, and 0.8 mgC $L^{-1}$; sd, 1.0 mgC $L^{-1}$ in
groundwater discharge. The same ratio of DOC reactivity was applied for three classes of POC
degradability. The kinetics for POC and DOC hydrolysis and parameters however are different
(Billen and Servais, 1989; Garnier et al., 2002).
DIC and TA are brought by subsurface and groundwater discharges (Venkiteswaran et al.,
2014). DIC is defined by the sum of bicarbonates ($HCO_3^-$), carbonates ($CO_3^-$) and $CO_2$. Unlike
$HCO_3^-$ and $CO_3^-$ measured in groundwater on a regular basis by French authorities (ADES,
www.ades.eaufrance.fr, last accessed 2020/02/11), $CO_2$ concentrations were not measured in
their survey. TA values are also provided in the ADES database.
To calculate DIC concentrations in groundwaters, we therefore used our own $CO_2$
measurements, equaling on average 15.92 mg C $L^{-1}$; sd, 7.12 mgC $L^{-1}$ (55 measurements in six
piezometers in the Brie aquifer during 2016–2017) (see methodology in Marescaux et al.,
2018a). DIC and TA were averaged for the 48 unconfined hydrogeological MESO units of the
basin (see concentrations in S4) during the recent period (2010–2015), including the simulation
period. In Figure 3, a summary of TA and DIC inputs by MESO units is shown by grouping
MESO units according to lithology and geological ages.

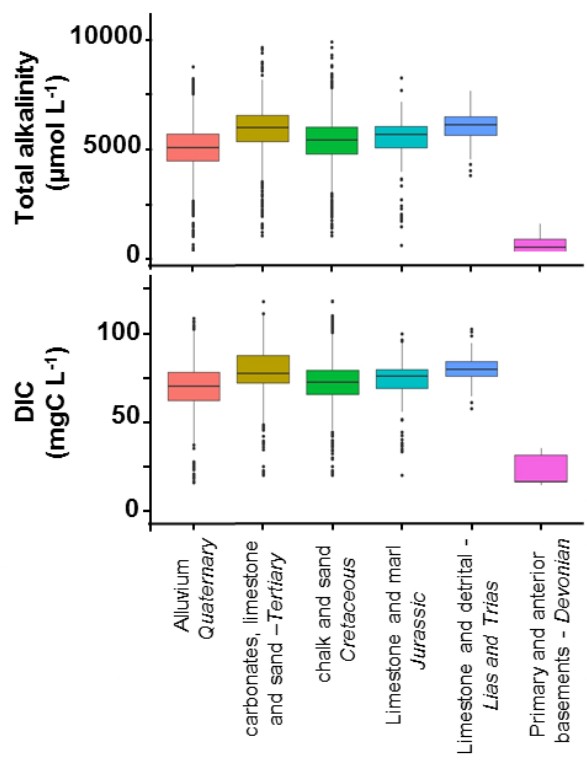


*Figure 3 Boxplots of total alkalinity (µmol L⁻¹) and dissolved inorganic carbon (DIC, mgC L⁻¹) groundwater concentrations by grouping the MESO units. The lower, intermediate and upper parts of the boxes represent, respectively, the 25th, 50th and 75th percentiles and the circles represent the outlier values (source: ADES). The color code is the same as the one in S4 spatially representing the MESO units of the basin.*

Documenting TA and DIC diffuse sources based on MESO units ensures a representation of their spatial heterogeneity in the Seine River basin. Carbonate waters showed higher TA and DIC mean concentrations while crystalline waters had the lowest mean concentrations in TA and DIC (primary and anterior basements from Devonian, Figure 3). Aquifers from Tertiary and alluvium from Quaternary had a more heterogeneous distribution of their concentrations (Figure 3). TA and DIC by MESO units were then spatially averaged at the scale of each modeling unit of the pyNuts-Riverstrahler model (69 modeling units, subdivided according to

Strahler ordination, S2), thus forming a semi-distributed estimate of groundwater
concentrations.
TA and DIC measurements in lower order streams cannot be considered as representative of
subsurface concentrations because lower order streams are expected to degas strongly in a few
hundred meters, as shown for $N_2O$ by Garnier et al. (2009) and for $CO_2$ in Öquist et al. (2009).
We have considered similar concentrations and spatial distribution for subsurface components
to those obtained for groundwater (from 25 to 92 mgC $L^{-1}$ DIC, and from 663 to 5580 µmol $L^{-1}$
TA, Figure 3).

**Point sources from WWTP effluents**

The pyNuts-Riverstrahler model integrates carbon and nutrient raw emissions from the local
population starting from the collection of household emissions into sewage networks until their
release after specific treatments in WWTPs. In the Seine River basin, most of these releases are
adequately treated before being discharged to the drainage network. DOC discharge from
WWTPs was described according to treatment type, ranging from 2.9 to 9.4 gC $inhab^{-1}$ $day^{-1}$
while POC discharge ranged from 0.9 to 24 gC $inhab^{-1}$ $day^{-1}$ based on the sample of water
purification treatment observed in the Seine basin (Garnier et al., 2006; Servais et al., 1999).
TA and DIC were measured at eight WWTPs selected to reflect various treatment capacities
(from 6 $10^3$ inhab eq to 6 $10^6$ inhab eq) and different treatment types (activated, sludge,
Biostyr® Biological Aerated Filter) in the Seine River basin. Sampling and analysis protocols
are provided in S5. This sampling did not allow us to highlight differences in per capita TA and
DIC emissions. Consequently, we used a fixed value of 3993 µmol $L^{-1}$ for TA and 70 mgC $L^{-1}$
for DIC, which correspond to the weighted mean by WWTP capacity of our measurements and
are in agreement with values from Alshboul et al. (2016) found in the literature.

**Impact of the reservoirs**


Nutrients and organic carbon cycling within the three reservoirs of the Seine River network
were simulated using the biogeochemical RIVE model adapted for stagnant aquatic systems
(Garnier et al., 1999). Owing to the absence of an inorganic carbon module in the modeling of
reservoirs yet, we used mean measurements of TA and DIC in reservoirs as forcing variables
to the river network. The Der lake reservoir was sampled three times (2016/05/24, 2016/09/12,
2017/03/16) and among others, TA and DIC were measured (see Table 3). Recent sampling
campaigns showed that TA and DIC are similar for the three reservoirs (X. Yan, pers. comm.).

*Table 2 Summary of the carbon related inputs of the pyNuts-Riverstrahler model.*

| Input variables | Flow | Database | averaged | values | source |
|---|---|---|---|---|---|
| DOC | subsurface | AESN | land use | mean: 3.13 mgC L$^{-1}$; sd: 4.56 mgC L$^{-1}$; | http://www.eau-seine-normandie.fr/ |
|  | groundwater | ADES | MESO units | mean: 0.91 mgC L$^{-1}$; sd: 0.8 mgC L$^{-1}$ | www.ades.eaufrance.fr |
| POC | subsurface | LUCAS, BioSoil and SoilTrEC Projects | based on estimated total suspended solids (TSS) fluxes, associated with a soil organic carbon (SOC) content | mean: 8.2 mgC L$^{-1}$, sd: 10.4 mgC L$^{-1}$ | (Aksoy et al., 2016) |
|  | groundwater |  |  | mean: 0.8 mgC L$^{-1}$, sd: 1.0 mgC L$^{-1}$ |  |
| DIC | subsurface | ADES | MESO units | from 25 to 92 mgC L$^{-1}$ | www.ades.eaufrance.fr |
|  | groundwater |  |  | from 25 to 92 mgC L$^{-1}$ |  |
| TA | subsurface | ADES | MESO units | from 663 to 5580 µmol L$^{-1}$ | www.ades.eaufrance.fr |
|  | groundwater |  |  | from 663 to 5580 µmol L$^{-1}$ |  |
|  |  |  |  |  |  |
| DOC | Point sources | Measurements | According to WWTP treatment and capacity | 2.9 to 9.4 gC inhab$^{-1}$ day$^{-1}$ | (Garnier et al. 2006; Servais et al. 1999) |
| POC | Point sources | Measurements |  | 0.9 to 24 gC inhab$^{-1}$ day$^{-1}$ |  |
| DIC | Point sources | Measurements | weighted mean by WWTP capacity | 70 mgC L$^{-1}$ | This study |
| TA | Point sources | Measurements | weighted mean by WWTP capacity | 3993 µmol L$^{-1}$ | This study |
|  |  |  |  |  |  |
| DIC | Reservoirs | Measurements in the Der Lake | by year | mean: 23 mgC L$^{-1}$, sd: 4 mgC L$^{-1}$ | This study |
| TA | Reservoirs | Measurements in the Der Lake | by year | mean: 1890 µmol L$^{-1}$, sd: 350 µmol L$^{-1}$ | This study |

## 2.2.3.   Observational data

We selected the 2010–2013 timeframe for setting up and validating the new inorganic module.

This period includes the year 2011, which was particularly dry in summer (mean annual water

discharge at Poses, 366 m$^3$ s$^{-1}$) and 2013, which was wet (mean annual average water discharge

at Poses, 717 m$^3$ s$^{-1}$) while 2010 and 2012 showed intermediate hydrological conditions (mean

annual average water discharges at Poses, 418 m$^3$ s$^{-1}$ and 458, m$^3$ s$^{-1}$, respectively) (data source:

Banque Hydro).

The pCO$_2$ values (ppmv) were calculated using CO2SYS software algorithms (version 25b06,

Pierrot et al., 2006) based on existing data collected by the AESN. TA, pH, and water

temperature data sets were used for the 2010–2013 selected period (8693 records for these three
variables, i.e., around 1209 stations distributed throughout the Seine basin, measurements that
were taken at a fixed time – 9:00-15:00 UTC–, and could not represent diurnal fluctuations).
The carbonate dissociation constants (K1 and K2) applied were calculated from Millero (1979)
with zero salinity and depending on the water temperature. Because $pCO_2$ calculations from pH
and TA can lead to overestimation of $pCO_2$ (Abril et al., 2015), the $pCO_2$ calculated data were
corrected by a relationship established for the Seine River and based on $pCO_2$ field
measurements (Marescaux et al., 2018b). To compute the interannual average over the 2010–
2013 period, data were averaged monthly, then annually at each measurement station and then
spatially averaged (i.e., by Strahler orders). Four stations offering sufficient data for the 2010–
2013 period were selected for appraising seasonal patterns. They are located along the main
stem of the Marne-Lower Seine River: Poses (the outlet), Poissy (downstream of the SAV
WWTP), Paris and Ferté-sous-Jouarre (upstream of Paris) (Figure 1a).
All data were processed using R (R Core team, 2015) and QGIS (QGIS Development Team,
2016). Kruskal-Wallis tests were used to compare simulated and measured $pCO_2$ averages.

### 2.2.4.    Evaluation of the model

Root mean square errors normalized to the range of the observed data (NRMSE) were used to
evaluate the pyNuts-Riverstrahler model including the inorganic module, indicating the
variability of the model results with respect to the observations, normalized to the variability of
the observations. NRMSE analysis were performed on inter-annual variations per decade for
the 2010-2013 period, combining observations and simulations at four main monitoring stations
along the longitudinal profile of the Seine River: Poses, Poissy (downstream of Paris), Paris,
and Ferté-sous-Jouarre (upstream of Paris).

# 3. Results

## 3.1. Simulations of spatial and seasonal variations of pCO₂.

### 3.1.1. CO₂ from lower order streams to larger sections of the Seine River

Simulations of $CO_2$ concentrations averaged for 2010–2013 by Strahler orders showed that pyNuts-Riverstrahler succeeded in reproducing the general trends of $CO_2$ observations (7565 data) (Figure 4). Although differences in $CO_2$ concentrations between the different order streams were not significant, their means tended to decrease from lower order streams (SO1) (width < 100 m) to SO5, and to finally increase in the higher order streams (width > 100 m) from SO6 to SO7, downstream of the Paris conurbation. Some discrepancy appeared for order 1, with simulations yielding higher values than the observations while for orders 2–7 simulation values were conversely lower than observation values. The corresponding $k$-values calculated for the Seine ranged from 0.04 to 0.23 m h⁻¹ with higher values in the first streams and lower values in larger rivers (not shown), with $CO_2$ outgassing positively related to the $k$-value (S3.5 Eq. S25).

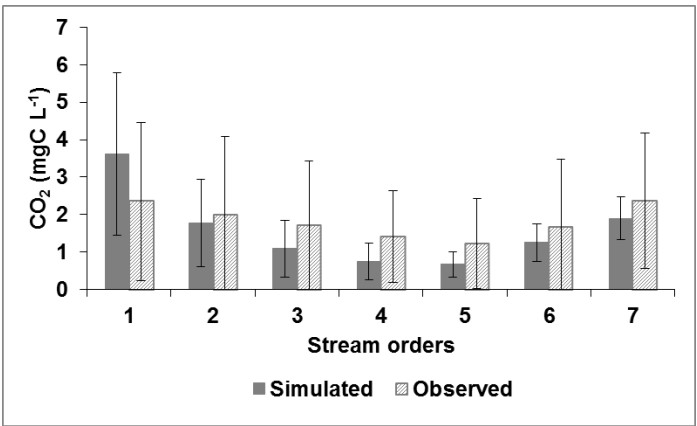

*Figure 4 Carbon dioxide concentrations in the Seine waters (CO₂, mgC L⁻¹) simulated by the pyNuts-Riverstrahler model (dark gray) and observed (light grey) as a function of the stream order averaged over the 2010–2013 period (whiskers indicating standard deviations).*

### 3.1.2.  Profiles of the main stem Marne and Lower Seine (at Poses)

In the same period (2010-2013), a focus on the main stem from the Marne River (SO6) until the outlet of the Seine River (Poses, SO7) showed that the model correctly reproduced longitudinal variations. Higher concentrations of $CO_2$ downstream of Paris, and a peak of $CO_2$ concentrations immediately downstream of the SAV WWTP were followed by a progressive decrease until the estuary (Figure 5). Note that the estuarine $CO_2$ concentrations were specifically modeled by Laruelle et al. (2019), using these outputs of the Riverstrahler simulations.

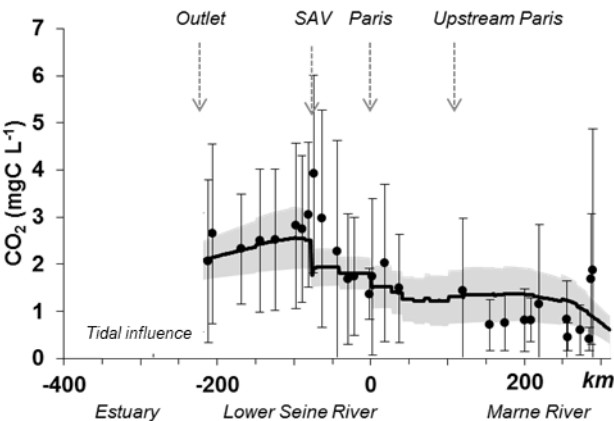

*Figure 5 Observed (dots) and simulated (line) mean carbon dioxide concentrations ($CO_2$, mgC $L^{-1}$) along the main stem of the Marne River (km −350 to 0) and the lower Seine River (km 0–350) averaged over the 2010–2013 period. The simulation envelope (gray area) represents standard deviations of simulated $CO_2$ concentrations. Whiskers are standard deviations between observed $CO_2$ concentrations.*

### 3.1.3.  Seasonal variations

Upstream, within Paris, and downstream of Paris, the model provides simulations in the right order of magnitude of the observed $CO_2$, DIC, TA and pH values, despite the fact that TA was underestimated in the two upstream stations selected for all seasons (Figure 6). DIC and TA

simulations followed the observed seasonal patterns with a depletion of concentrations
occurring in summer/autumn related to low-flow support by the reservoirs. Indeed, reservoirs
showed lower TA and DIC concentrations than rivers (Table 3). In addition to the intra-/inter-
stream order variabilities of $CO_2$ (Figure 4), $CO_2$ concentrations showed a wide spread in values
over the year (Figure 6). Although simulated $CO_2$ concentrations fitted rather well with the
level of the observations (NRMSE = 15%), the model tended to overestimate the winter values
upstream and within Paris (Figure 6, left).
For DIC, simulations upstream from Paris (Figure 6, right) seemed lower than the observations
(but summer data are missing); however, downstream at the other three stations selected,
simulations accurately represented the observations (Figure 6, NRMSE = 15%). Seasonal
variations of TA were satisfactorily reproduced by the simulations, although they were slightly
underestimated by the model at the stations upstream and downstream of Paris (Figure 6,
NRMSE = 25%). Regarding pH, simulations were in a similar range as the observations (range,
7.5–8.5), and lower summer pH values in the lower Seine were correctly simulated by the model
(Figure 6, NRMSE = 17%).

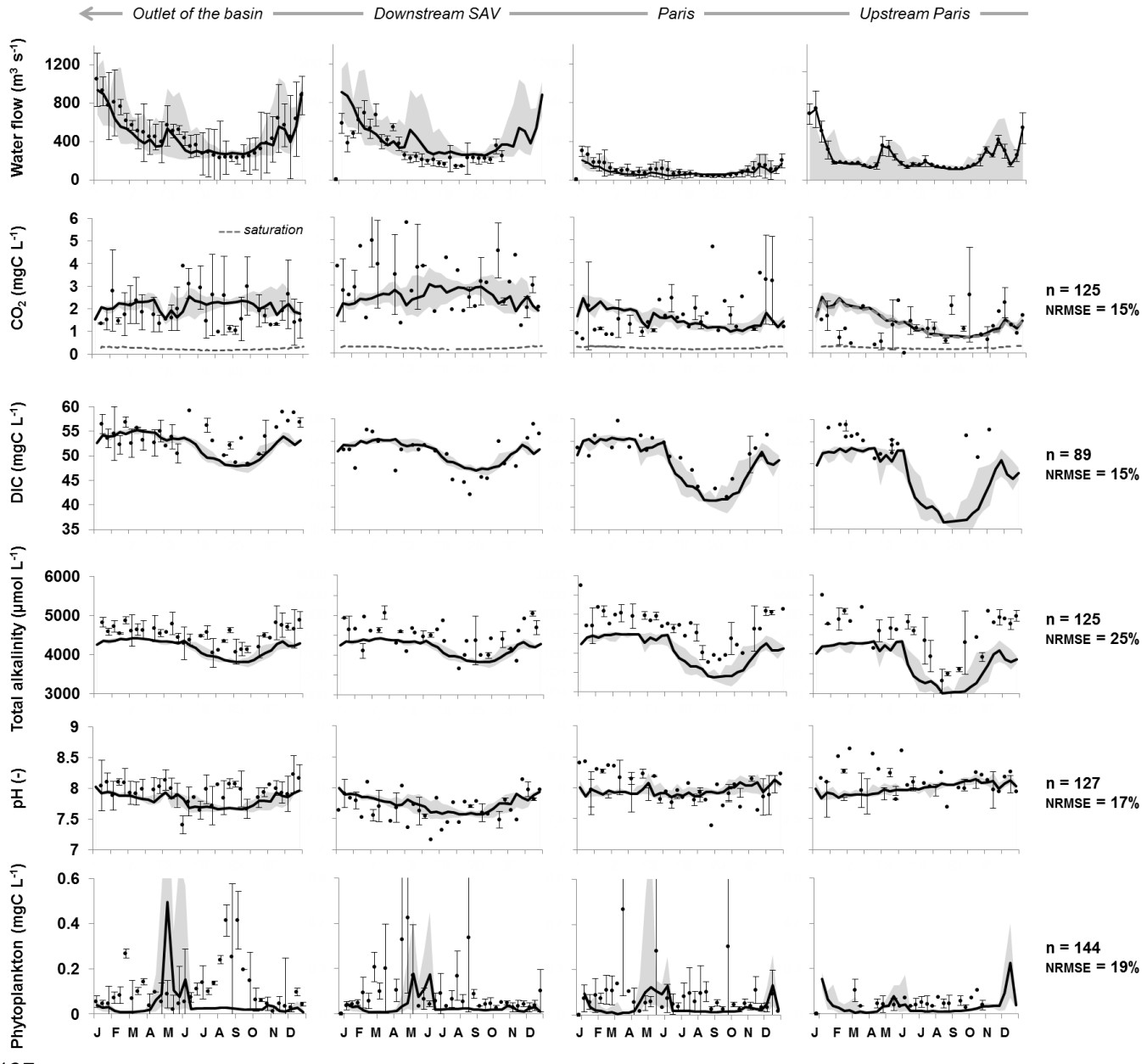


*Figure 6 Ten-day simulated (lines) and observed (dots) water discharges over the 2010–2013 period (Q, m³ s⁻¹),*

*concentrations of carbon dioxide (CO₂, mgC L⁻¹, and CO₂ sat, mgC L⁻¹), dissolved inorganic carbon (DIC, mgC*

*L⁻¹), total alkalinity (TA, µmol L⁻¹), pH (-), and phytoplankton (mgC L⁻¹). Four monitoring stations of interest*

*along the main stem of Marne-lower Seine are shown: Ferté-sous-Jouarre (upstream of Paris on the Marne*

*River), Paris on the lower Seine (upstream at Charenton), downstream of the SAV WWTP, and at the outlet of*

*the basin (Poses). NRMSE analysis were performed on inter-annual variations per decade for the 2010-2013*

*period, combining observations and simulations at four main monitoring stations. Simulation envelope*

*corresponds to standard deviations (gray area). For observed data, whiskers are standard deviations.*

Although the level of phytoplankton biomass was adequately simulated, the summer bloom
observed at the outlet was not reproduced, whereas the early spring bloom observed in the lower
Seine was simulated with a time lag compared to the observations (Figure 6, bottom, NRMSE
= 19%).

### 3.1.4. Selection of a gas transfer velocity

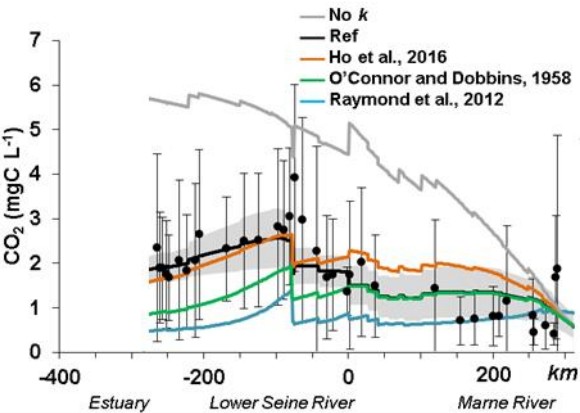


*Figure 7 Influence of the gas transfer velocity formalisms along the main stem of the Seine River basin (Marne –*

*Lower Seine River) impacted riverine $CO_2$ concentrations.*

The way of taking into account the gas transfer velocity in the modeling approach could explain
these discrepancies in SO6 and SO7 (Figure 4). Different values of $k$ were explored specifically
in the downstream part of the Seine river network (SO6 and SO7 where river width exceeds
100m) (Figure 7). Indeed, the gas transfer velocity value reported by Alin et al. (2011) was used
for streams and rivers up to 100 m wide, as they recommended. Whereas these $k$-values
provided adequate simulations in the river up to 100 m wide, for river widths greater than 100
m, we tested different $k$-values. In larger stream orders, we showed that calculations of $k$
according to the Equation 5 of Table 2 by Raymond et al. (2012), induced a too high outgassing
while when not using any $k$-value for these larger rivers, the opposite behavior with a much too
low outgassing of $CO_2$ was observed.
Therefore, for river widths greater than 100 m, a $k_{600}$ equation based on O'Connor and Dobbins,
(1958) and Ho et al. (2016), neglecting the term related to the wind, and providing the most
accurate $CO_2$ concentrations, was selected (see S3 for more information's on the selection of $k$
and the tests performed):
Although these results can be improved, organic and inorganic carbon and total alkalinity
budgets can be calculated at the scale of a whole drainage basin for the first time.

## 3.2. Alkalinity, inorganic and organic carbon budgets

We established an average inorganic and organic budget for the period studied (2010–2013)
(Table 4). The budget of inorganic and organic carbon (IC and OC) of the entire Seine River
basin (from headwater streams to the beginning of the estuary) showed the high contribution of
external inputs (sum of point and diffuse sources accounted for 92% and 68% of IC and OC
inputs, respectively) and riverine exports (68% and 66% of IC and OC outputs, respectively).
These exports were at least one order of magnitude higher for the IC budget (Table 4). The
substantial contribution of the Seine aquifer water flow led the IC flux brought by groundwater
to dominate over those from the subsurface (respectively, 57.5% vs. 34% of total IC inputs,
respectively), while for OC, the subsurface contributions were higher than the groundwater
contributions (54% vs. 14% of the total OC fluxes).
Interestingly, the relative contributions of point sources to OC inputs were higher than for IC
(23% and 7% of the OC and IC inputs, respectively) (Table 4).
Heterotrophic respiration by microorganisms accounted for only 1.5% of the IC inputs.
Similarly, IC losses by net primary production also accounted for a small proportion, i.e., 0.6%
of the IC inputs. For the OC budget, despite a contribution of autochthonous inputs from
instream biological metabolisms (NPP and nitrification, 9% of inputs, and heterotrophic
respiration, 7%), which was relatively high compared with their proportion in IC fluxes (2.3%),
allochthonous terrestrial inputs still dominated the OC budget (Table 4).
The Seine River, at the outlet, exported 68% of the IC entering or produced in the drainage
network, and 66% of the OC brought to the river (including both particulate and dissolved
forms) (Table 4). Instream OC losses were related to heterotrophic respiration (7%) and to a
net transfer to the benthic sediment compartment, including sedimentation and erosion
processes (estimated at 28% of losses). In the IC budget, $CO_2$ emissions were a substantial
physical process (31% of the overall losses) (Table 4).
A similar calculation was performed for the total alkalinity (TA) budget. As for inorganic
carbon, the contribution of internal processes remained relatively low compared with the high
levels of TA in lateral inputs (diffuse sources: 93 %; point sources: 6 %) and flows exported to
the basin outlet (97 %). Indeed, instream production mostly relied on heterotrophic respiration
(< 1%) while denitrification was negligible. Photosynthesis might also produce or consume
alkalinity whether $NO_3^-$ or $NH_4^+$ is the preferential N source of phytoplankton's uptake, but in
our budget it resulted in our budget in a net TA reduction (2%), while nitrification also
contributed to less than 1% of TA output.
*Table 4 Budget of the Seine hydrosystem for inorganic and organic carbon (kgC km$^{-2}$ yr$^{-1}$) and total alkalinity*
*(TA, mol km$^{-2}$ yr$^{-1}$) as calculated by the pyNuts-Riverstrahler model averaged over the period 2010-2013. \* TA*
*input related to NPP refers to the net difference between TA produced by photosynthesis on NO$_3$ uptake and*
*photosynthesis on NH$_4$ uptake (reducing alkalinity). \*\*Net sediment loss is the difference between the erosion*
*and the sedimentation calculated by the model.*

| 2010-2013 | **Processes involved in inorg C budget** | kgC km$^{-2}$ yr$^{-1}$ | % |
|---|---|---|---|
| **Input to river** | Diffuse sources from subroot | 5963 | 34.4 |
| | Diffuse sources from groundwater | 9968 | 57.5 |
| | Urban point sources | 1135 | 6.6 |
| | Heterotrophic respiration | 266 | 1.5 |
| | Denitrification | 0 | 0.0 |
| **Output from river** | Delivery to the outlet | 12483 | 68.4 |
| | CO$_2$ emissions | 5619 | 30.8 |
| | Nitrification | 37 | 0.2 |
| | NPP | 105 | 0.6 |
| | | | |
| 2010-2013 | **Processes involved in TA budget** | mol km$^{-2}$ yr$^{-1}$ | % |
| **Input to river** | Diffuse sources from subroot | 360983 | *34.9* |
| | Diffuse sources from groundwater | 604145 | *58.4* |
| | Urban point sources | 66770 | *6.4* |
| | Heterotrophic respiration | 2972 | *0.3* |
| | Denitrification | 0 | *0.0* |
| **Output from river** | Delivery to outlet | 1004299 | *97.1* |
| | Nitrification | 6219 | *0.6* |
| | NPP \* | 24352 | *2.4* |
| | | | |
| 2010-2013 | **Processes involved in org C budget** | kgC km$^{-2}$ yr$^{-1}$ | % |
| **Input to river** | Diffuse sources from subroot | 870 | 53.9 |
| | Diffuse sources from groundwater | 227 | 14.1 |
| | Urban point sources | 375 | 23.2 |
| | Nitrification | 37 | 2.3 |
| | NPP | 105 | 6.5 |
| **Output from river** | Delivery to the outlet | 1086 | 65.7 |
| | Heterotrophic respiration | 110 | 6.7 |
| | Net sedimentation \*\* | 456 | 27.6 |

## 3.3. Carbon aquatic processes

Whereas IC and OC budgets of the Seine hydrosystem were clearly dominated by external terrestrial inputs and outputs through deliveries at the coast, an attempt was made here to analyze instream processes involved in the IC and OC cycles (Figure 8, Figure 9).

The average spatial distribution of IC processes, as calculated by the model, was mapped for the 2010–2013 period (Figure 8). Benthic activities were the greatest in smaller streams. By contrast, net primary production and heterotrophic planktonic respiration, which both followed a similar spatial pattern, increased as Strahler order increased, reaching their highest values in the lower Seine River. All these biological processes involved in the IC cycle were therefore highly active in the main stem of the river, while on the other hand $CO_2$ outgassing occurred mainly in the basin's small headwater streams (Figure 8).

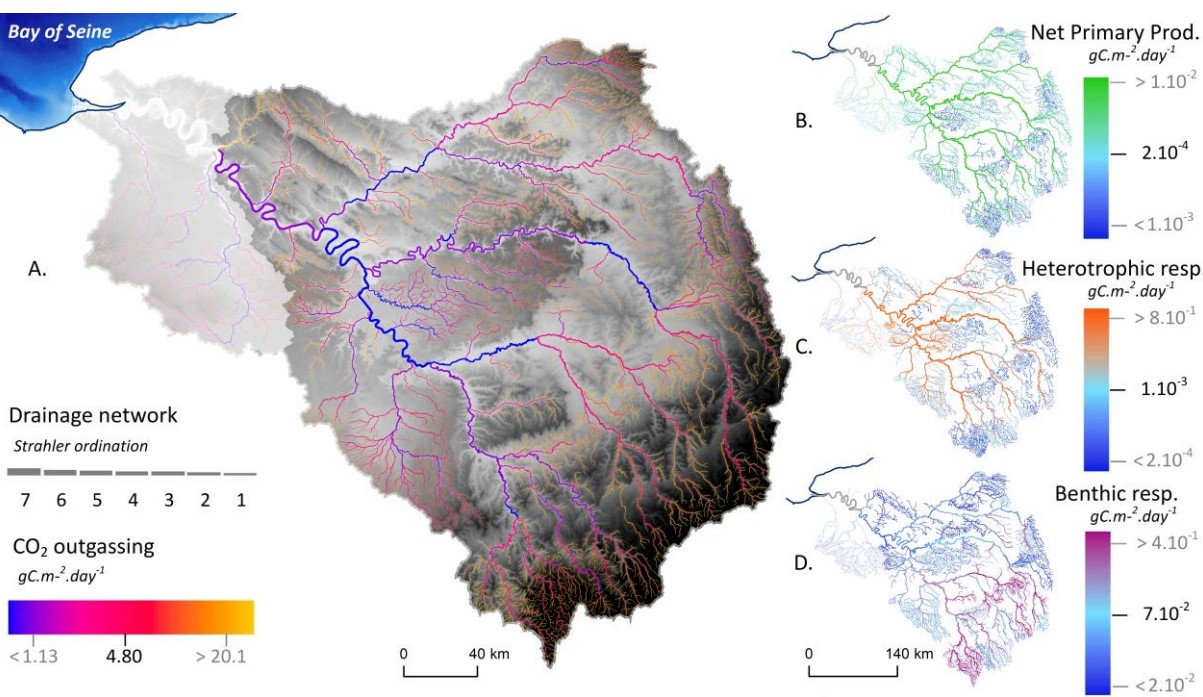

*Figure 8 Instream processes involved in the inorganic carbon cycle simulated by pyNuts-Riverstrahler and averaged over the 2010–2013 period for the Seine River network until its fluvial outlet at Poses. a) $CO_2$ outgassing (blue–yellow, gC $m^{-2}$ $day^{-1}$); b) net primary*

*production (blue–green, gC m$^{-2}$ day$^{-1}$); c) heterotrophic planktonic (blue–violet); d) benthic*
*respiration (blue–orange, gC m$^{-2}$ day$^{-1}$) are represented in the hydrographic network.*
Regarding the OC processes, mostly linked to biological activity, they were analyzed in terms
of ecosystem metabolism (Figure 9). The net ecosystem production (NEP, gC m$^{-2}$ day$^{-1}$) is
defined as:
$$NEP = NPP - Het.\ Respiration$$
where NPP is the net primary production (gC m$^{-2}$ day$^{-1}$) depending on the growth of
phytoplankton. NPP contributes to building phytoplankton biomass that constitutes a stock of
organic carbon, emitted in turn as $CO_2$ by respiration (Het. respiration, gC m$^{-2}$ day$^{-1}$).
Simulations showed that NEP would remain negative in the entire drainage network (Figure 9).
However, NEP must be analyzed with caution since the phytoplankton pattern was not
adequately represented (see Figure 6). In SO1, this negative NEP was associated with almost
no NPP, and heterotrophic respiration was dominated by benthic activities (see Figure 8). In
SO5, NEP was less negative than in SO1 (Figure 9), and heterotrophic respiration was lower
than in SO1 while NPP was higher. In the lower Seine River (SO7), NPP increased as did
heterotrophic respiration, which reached its highest value in this downstream stretch receiving
treated effluents from WWTPs. Therefore, the increase in NPP did not result in positive NEP.
The entire drainage network was thus supersaturated in $CO_2$ with respect to atmospheric
concentrations, and constituted a source of $CO_2$. This supersaturation was the highest in smaller
orders, lower in intermediate orders and increased again in the lower Seine River (Figure 4, see
also Figure 8).

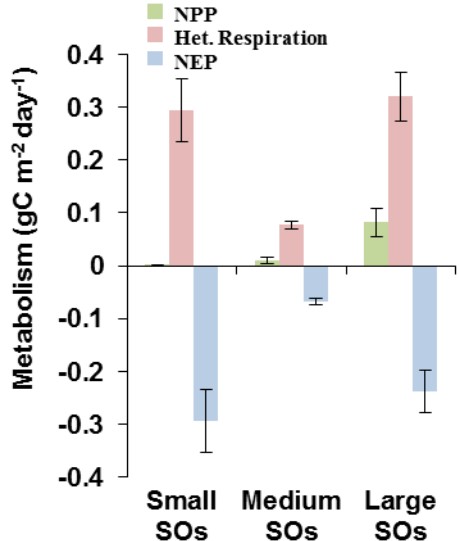


*Figure 9 Metabolism for small, intermediate and large stream orders (SO) (here represented*

*by SO1, SO5, and SO7, respectively) of the Seine basin simulated by pyNuts-Riverstrahler*

*and averaged over the 2010–2013 period. Net primary production (NPP, gC $m^2$ $day^{-1}$),*

*heterotrophic respiration (Het. respiration, gC $m^2$ $day^{-1}$), net ecosystem production (NEP, gC*

*$m^2$ $day^{-1}$).*

## 4. Discussion

## 4.1. Evaluation of the model

Simulated $CO_2$ concentrations tend to be higher than observed ones for SO1. These differences

may be related to the high variability of $CO_2$ in SO1, and the scarcity of measurements in spring.

However, Öquist et al. (2009) estimated that up to 90% of daily soil DIC import into streams

was emitted to the atmosphere within 200 m. Such a $CO_2$ emission pattern can be applied to the

Seine, as a similar result was found for $N_2O$ (Garnier et al., 2009). Since soil emissions were

very difficult to capture, we considered that concentrations in groundwater (DIC and TA)

closely reflect the composition of diffuse sources, much like soil composition. This assumption

probably underestimates the DIC/TA ratio brought to the river in lower order streams.

Differently from SO1, simulated concentrations in SO2–7 are lower than the observed values
(Figure 4). Overall, the NRMSE indicating a percentage of variation was less than 20%, except
for TA (25%).
Regarding gas transfer velocity values, an equation for large rivers with no tidal influence using
wind speed could be more appropriate (Alin et al., 2011) and could decrease NRMSE in these
downstream sections of the river. However, the Riverstrahler model does not consider wind as
an input variable, which would have required the model to have a much higher spatiotemporal
resolution to reflect its spatiotemporal heterogeneity in the Seine basin, with for example, the
diurnal cycle affected by phenomena such as breezes (Quintana-Seguí et al., 2008).
Future work with direct $k$ measurements and/or a new representation of $k$-values in the model
could help improve outgassing simulations with pyNuts-Riverstrahler. A test of different $k$
formulations on high stream orders (width > 100 m) representing only 1.5% of the length of the
river system showed an increase of the total $CO_2$ outgassing estimates by up to 6.2%. Our model
is $k$ sensitive and our estimates differs from the results of Lauerwald et al. (2017), who observed
that a large variation in $k$ does not lead to a significant change in simulated aquatic $CO_2$
emissions. For the Seine River here, we indeed used a more accurate $k$-value calculated at each
time step (10 days) and at every kilometers of the river network (according to water
temperature, velocity, depth). In addition, a huge organic carbon load is brought by WWTPs in
this Seine urbanized hydrosystem that disrupts carbon dynamics (e.g., WWTPs treating 12
million inhab. eq in the Parisian conurbation) in the downstream part of the Seine River, in
contrast to simulations on a natural network (Lauerwald et al., 2017).

Regarding seasonal patterns, DIC and alkalinity amplitudes were suitably captured and the level
of the values was correct. DIC and TA observations showed a strong decrease from June/July
to November (maximum amplitude decrease, 10 mgC $L^{-1}$ and 1000 µmol $L^{-1}$), as illustrated by
the model. For the Seine River, the water flow decrease in summer was mainly related to the
decrease in runoff water, meaning that the groundwater contribution was comparatively higher
at this time. According to our measurements, these groundwaters were more concentrated in
TA, DIC, and $CO_2$ than runoff water. However, water released by upstream reservoirs
(supporting low flow in the downstream section of the Seine network) account for a significant
proportion of the river discharge during summer and was characterized by lower TA, DIC and
$CO_2$ concentrations. Then the decrease observed was related to the contribution of reservoirs.
These results strongly encourage the implementation of an inorganic carbon module in the
modeling of reservoirs, already coupled with Riverstrahler for nutrients and organic carbon
(Garnier et al., 1999).

The model showed a weak performance in representing $CO_2$ seasonality. Referring to a previous
study (Marescaux et al., 2018b), $pCO_2$ seasonality in the Seine River resulted from a
combination of water temperature and hydrology leading to an increase in $pCO_2$ and $CO_2$
evasion fluxes from winter to summer/autumn. The pyNuts-Riverstrahler model however has
an accurate representation of these constraints and would not account for these discrepancies.
Also, despite the fact that the biomass level of phytoplankton was consistent with the
observations, the seasonal pattern was not satisfactory reproduced by the model. However, it is
worth mentioning that phytoplankton parameters in RIVE were determined through laboratory
experiments at a time when the amplitude of algal blooms was much higher than at present (up
to 4.5-6 mgC $L^{-1}$ i.e., chlorophyll *a* reaching 150 µgChla $L^{-1}$, Garnier et al., 1995). Indeed, the
implementation of the European Water Framework Directive in the 2000s with enhancement
of treatments in WWTPs greatly improved water quality (Romero et al., 2016). New laboratory
experiments for possibly taking into account additional phytoplankton groups or species in
these new trophic conditions and/or mixing stochastic and mechanistic modeling are required
to better represent phytoplankton temporal dynamics in the model. In addition, the observed
incident light, instead of the empirical relationship used, would improve the early winter bloom,
newly occurring in a changing environment.

## 4.2. Export fluxes

The new implementation of an inorganic carbon module in the pyNuts-Riverstrahler model
allows us to estimate $CO_2$ outgassing of the Seine River at $364 \pm 99$ GgC yr$^{-1}$ ) (1.4 GgC km$^{-2}$
yr$^{-1}$ taking into account a river surface area of 260 km$^2$). This is significantly lower than in our
previous estimate of 590 GgC yr$^{-1}$ (2.2 GgC km$^{-2}$ yr$^{-1}$ from a river surface area of 265 km$^2$)
using $CO_2$ measurements only (Marescaux et al., 2018a). This difference is explained by
various factors. Marescaux et al (2018a) use *k* formulates according to Raymond et al. (2012,
Eq. 5 in Table 2) all along the Seine drainage network and consequently, the value of $CO_2$
emissions was most likely overestimated (see 4.1. Evaluation of the model). We also
acknowledged that the $CO_2$ outgassing estimate yielded by simulations might overall slightly
underestimate emissions with respect to Figure 4, which showed that our simulated $CO_2$
concentrations were overestimated for SO1 but underestimated for SO2 to SO7. In the model,
a better spatio-temporal resolution and description of the water temperature, the water velocity
and a more accurate description of the *k*-value adopted here with different *k*-values for small
and high stream orders would be associated with less outgassing than in our previous study. For
this reasons, we believe that our estimate of $364 \pm 99$ GgC/yr, using our process based model
is a more accurate value of $CO_2$ emissions from the Seine River.
The outgassing found for the Seine River by surface area of river of $1400 \pm 381$ gC m$^{-2}$ yr$^{-1}$ is
in the middle range of the average estimates of outgassing from temperate rivers (70-2370 gC
m$^{-2}$ yr$^{-1}$), including the St. Lawrence River (Yang et al., 1996), Ottawa River (Telmer and
Veizer, 1999), Hudson River (Raymond et al., 1997), US temperate rivers (Butman and
Raymond, 2011) and Mississippi River (Dubois et al., 2010). This high variability for these
temperate rivers is strongly dependent on whether or not the first-order streams were considered
in the outgassing. Similar to our study, Butman and Raymond (2011) took into account lower
order streams and rivers while lower estimates correspond to studies investigating large rivers,
excluding lower order streams. Indeed, outgassing are often greater in headwater streams than
in large rivers owing to higher $CO_2$ concentrations and headwater streams have higher gas
transfer velocities (Marx et al., 2017; Raymond et al., 2012a). The mapping of $CO_2$ outgassing
in the Seine basin clearly showed these spatial trends, with smaller streams releasing more $CO_2$
than median and larger rivers (see Figure 8). Indeed, first-order streams of the Seine River
represents 9.6% of the Seine surface area and contributed to 40% of the total $CO_2$ emissions by
the river network.
Regarding organic carbon, Meybeck (1993) estimated the DOC export to the ocean for a
temperate climate at 1.5 gC m$^{-2}$ yr$^{-1}$, a value that is higher than our OC estimate of 1.1 gC m$^{-2}$
yr$^{-1}$ for the Seine River basin, before entering the estuarine section. Compared with other
temperate rivers, the rivers of the northern France, and specifically the Seine River here, are
rather flat, their low altitude limiting erosion (Guerrini et al., 1998). In addition, since the
implementation of the European Water Framework Directive in the 2000s, decreasing nutrients
and carbon in wastewater effluents discharged into the rivers (Rocher and Azimi, 2017),
together with a decrease in phytoplankton biomass development (Aissa Grouz et al., 2016;
Romero et al., 2016) can explain this difference in DOC fluxes for the Seine, a change probably
valid for many other western European rivers (Romero et al., 2013). Furthermore, the $CO_2/OC$
ratio of the export to the estuary of the Seine hydrosystem is 5.2, which is higher than this ratio
for the Mississippi River, for example (4.1; Dubois et al., 2010b; Li et al., 2013) and may be
related to considerable outgassing from headwater streams taken into account in our study.
Note, however, that the small Seine River basin exports only $70 \pm 99$ GgC $yr^{-1}$ OC compared
with the large Mississippi River with exports amounting to 2435 GgC $yr^{-1}$ OC (Dubois et al.,
2010), and with a surface area more than 40 times greater than the Seine. Interestingly, the
Seine River export was estimated at three times less than the export calculated in 1979 (250 Gg
C $yr^{-1}$ , Kempe, 1984). This difference in DOC concentrations in the Seine River would be 2.8
times lower than in the 1990s (Rocher and Azimi, 2017).
We estimated the DIC export of the Seine River at $820 \pm 220$ GgC $yr^{-1}$, a value higher than
basins of the same size or even larger (e.g., Ottawa River, drainage are, 149,000 $km^2$, 520 GgC
$yr^{-1}$ , Telmer and Veizer, (1999); Li et al. (2013)). The high concentrations of $HCO_3^-$ in the
Seine basin already documented and related to the lithology of the Seine basin (limestone and
gypsum beds from Cretaceous and Tertiary) (Kempe, 1982; 1984) may explain this high export
to the river outlet. With both high $CO_2$ and DIC exports, the ratio of $CO_2/DIC$ exports from the
Seine River is the same as the overall ratio here (0.5, Li et al., 2013)..

## 4.3. Metabolism

Model simulations with the new inorganic carbon module can be used to analyze spatial
variations of $CO_2$ in regard to instream metabolism activities. We observe that the influence of
the metabolism activities on the $CO_2$ outgassing is low. Indeed, in the carbonated Seine River,
the IC originating from groundwater supports the $CO_2$ outgassing along the network (Figure 8).
Nevertheless, instream metabolism activities produce or consume $CO_2$.
The model highlights the importance of benthic activities in headwater streams (Figure 8) that
decreased downstream as heterotrophic planktonic activities increased in larger rivers, a typical
pattern described by the river continuum concept (RCC, Vannote et al., 1980) and quantified
for the Seine River (Billen et al., 1994; Garnier et al., 1995; Garnier and Billen, 2007). These
results are also in agreement with those reported by Hotchkiss et al. (2015), who suggested that
the percentage of $CO_2$ emissions from metabolism increases with stream size while $CO_2$
emissions of lower-order streams are related to allochthonous terrestrial $CO_2$. Regarding
headwater streams, Battin et al. (2009b) described benthic activities as the highest (as also
observed in our study, Figure 8) where microbial biomass is associated with streambeds
characterized by exchanges with subsurface flow bringing nutrients and oxygen and increasing
mineralization.
Mean NEP would remain negative in the entire basin resulting from heterotrophic conditions
producing $CO_2$ (Figure 8 and Figure 9). However, even though the level of phytoplankton
biomass was correctly simulated, the summer downstream bloom, which was not reproduced
by the model, could lead to some NPP underestimation. As expected, NPP in lower order
streams was lower than in higher SOs owing to shorter water residence times. Benthic
respiration of lower order streams was significant (Figure 8) and made NEP highly negative.
Also, small SOs were the most concentrated in $CO_2$ owing to the groundwater contribution.
Intermediate stream orders showed the smallest $CO_2$ or heterotrophic respirations with NEP
less than $-0.1$ gC m$^{-2}$ day$^{-1}$. This can be explained by an increase of NPP due to a lower dilution
rate than the phytoplankton growth rate (Garnier et al., 1995), and to a reduced ratio of the
bottom sediment-to-water column volume, decreasing heterotrophic respiration. In higher
stream orders both NPP and heterotrophic respiration were the highest, however, they led to
negative NEP lower than SO1 (Figure 8 and Figure 9). Despite photosynthesis reducing the
$CO_2$ concentrations (Figure 6), the highest SOs were affected by wastewater effluents, resulting
in an overall negative NEP.
During the recent 2010–2013 period studied herein, and in all SOs, the NPP never exceeded
heterotrophic respiration (ratio NPP:Het.-Resp or P:R < 1) (Figure 9). Whereas in the past the
eutrophication of the Seine River led to a P:R ratio greater than 1 in large rivers, at least during
spring blooms, with P and R values increasing up to 2.5 gC $m^{-2}$ $day^{-1}$ (Garnier and Billen, 2007),
the P:R ratio is now systematically lesser than 1. These changes, linked to an overall decrease
in biological metabolism, are explained by improvements of treatments in WWTPs decreasing
the organic carbon load discharged into rivers and the associated pollution, and hence
decreasing the $CO_2$ concentration along the main stem of the Seine River (Marescaux et al.,
2018b). Beside DOC, improvements wastewater treatments also reduced nutrient inputs to the
river, especially phosphates, today a limiting nutrient to algal development in SO5 and 6,
reducing algal peaks by a factor of 3.

## 5. Conclusion

The pyNuts-Riverstrahler model of biogeochemical river functioning newly includes the
processes involved in the inorganic carbon cycle in order to represent the spatial dynamics and
seasonal variations of $CO_2$ concentrations and outgassing along the Seine hydrosystem. The
sensitivity of simulations to different gas transfer velocity values highlighted the need for
additional refinement for the Seine River so as to choose the best model equation. In addition,
revisiting the phytoplankton description in the model could facilitate a better simulation of the
temporal dynamics of phytoplankton. Further, an explicit representation of the anaerobic
reduction chain of the benthos could enable us to specify the benthic impact on TA and DIC in
a greater variety of ecosystems.
$CO_2$ concentrations appear to be controlled differently along the Seine hydrosystem. In small
orders, concentrations were mainly driven by diffuse sources. In larger rivers, in addition to the
influence of groundwater and low-flow support by upstream reservoirs, concentrations showed
patterns linked to hydrosystem metabolisms. Indeed, blooms tended to decrease $CO_2$
concentrations, although the hydrosystem remained heterotrophic and supersaturated with
respect to the atmospheric $CO_2$ concentrations. Heterotrophic respiration increased $CO_2$
concentrations with peaks downstream of WWTP effluents enriched in organic carbon.
Our Riverstrahler modeling has shown that there are many factors that control $CO_2$ emissions
in basins affected by human activity along an aquatic continuum. Once validated by field
measurements, which are still too scarce, this generic modeling approach can be applied to any
drainage system to better quantify lateral $CO_2$ emission on a continental scale.

## Data availability

The datasets generated during the current study are available from the corresponding author on reasonable request.

## Author contribution

All the authors contributed to the design of the study. J.G. and V.T. are co-supervisors of the PhD. A.M. participated as a PhD student in the field campaigns, lab chemical analyzes and implementation of the new inorganic carbon module. N.G. and M.S. provided technical and scientific support for the modelling. A.M. wrote the first draft of the manuscript, and all the co-authors helped to interpret the data and write the article.

## Competing interests statement

The authors declare no competing financial or non-financial interest.

## Acknowledgments

The project leading to this application received funding from the European Union's Horizon 2020 research and innovation program under the Marie Sklodowska-Curie grant agreement No. 643052. A PhD grant was attributed to Audrey Marescaux. Many thanks are due to Sébastien Bosc, Anunciacion Martinez Serrano and Benjamin Mercier for their kind participation in the fieldwork and for their assistance with chemical analyses in the lab. We thank Muriel Chagniot (Veolia Water, France), and the operators of the Veolia WWTPs for their precious help in organizing the field campaigns. The SIAAP (Vincent Rocher) is also sincerely acknowledged for their contribution to sampling the largest WWTP of the Paris conurbation and the long-term

view on treatments in the SIAAP WWTPs provided by their recent book (Rocher and Azimi,
2017). Vincent Thieu (assistant professor at the University Pierre and Marie Curie, Paris) and
Josette Garnier (Research Director at the Centre National de la Recherche Scientifique, France)
are co-supervisors of the PhD. Nathalie Gypens is Professor at the Université Libre de Bruxelles
(Belgium). Marie Silvestre is GIS Engineer at the Centre National de la Recherche Scientifique
(France).

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
