# Peer review of "Title page Modeling inorganic carbon dynamics in the Seine River continuum in France"

_Hydrology and Earth System Sciences, 2019_

## Referee Comment (RC1) · Anonymous Referee #1 · 13 Jan 2020

This paper described a biogeochemical model incorporating inorganic carbon cycle and applied the model to the Seine River system. The model was built based on an existing biogeochemical model and the model structure and setup have been sufficiently described. The results from current study help to fill up the gaps in understanding the contribution of inland waters to the global carbon cycle. However, the model performance is not very convincing. There had been a few other models able to simulate inorganic carbon in rivers and have not been discussed. In summary, the manuscript has potential to be improved and I would like to suggest the authors to consider:

1. improving the model performance presentation (see specific comments below);

2. the discussion of current findings is too site-specific; I would suggest to expand the discussion to a more general sense, e.g. how the inorganic carbon system in Seine

[Figure]

compared to other inland water systems? What are the meaning of current findings to estimating the roles of rivers in local and global carbon cycle? etc.

3. Also, the text writing in the introduction and discussion need to be polished. I list a few issues in the specific comments below but encourage the authors to go through the text and improve the writing in general.

Specific comments:

Line 37-38: some words are missing from this sentence. 'Outgassing was the most important {carbon sink/inorganic carbon process?}'

Line 69-71: This statement seems controversial to some other findings that eutrophic system usually contains richer organic matters and pCO2 (e.g. Borges and Abril, carbon dioxide and methane dynamics in estuaries, DOI: 10.1016/B978-0-12-374711-2.00504-0). Can you please explain more about this statement?

Line 69-71: The Xu et al. 2019 reference is missing;

Line 72-76: This statement needs to be treated carefully. Other methods, such isotope surveys, can also be used to investigate the fate of carbon in aquatic systems.

Line 85-90: A few early papers had reported models including the inorganic carbon cycle and pCO2 exchange but have not been mentioned here. Such as the CON-TRASTE model (Vanderborght et al 2002, Application of a transport-reaction model to the estimation of biogas fluxes in the Scheldt estuary, Biogeochemistry 59: 207-237), RTM model (Regnier et al 2013, modelling estuarine biogeochemical dynamics: from the local to the global scale, Aquat Geochem 19: 591-626); How is the current model compared to these models?

Line 91-92: This individual sentence as one paragraph is not reading well. Can be merged with next paragraphs.

Line 111: unit of the north and east coordinates?

Line 228-230: the gas transfer velocity only affect the exchange rate, not the change direction of $pCO_2$ (and therefore DIC).

Line 383-384: why only 4 years simulated but NRMSE were performed on inter-annual variations per decade, instead of 2010-2013? Also, normalized against mean observational data instead of inter-annual variations is more representative.

Line 402-404: as $CO_2$ concentrations are related to DIC and TA, it would be better if you show the comparisons of observed and modelled DIC/TA along with the $CO_2$ concentrations.

Line 517-518: can't find $CO_2$ outgassing in figure 9?

Figure 6: why there are two dark lines in the water flow of the outlet of the basin? Also, as the model timeframe includes dry and wet years, it is better to show the results year to year but not averaged from 4 simulated years;

Line 583-589: this sentence needs to be re-organized.

Line 624: left bracket is missing in citation;

Section 4.3: is there a relationship between the river eutrophic state and the metabolism activity, and $CO_2$ outgassing?

---

## Referee Comment (RC2) · Anonymous Referee #2 · 14 Jan 2020

The manuscript submitted by Marescaux et al. presents a technical upgrade of the pyNuts-Riverstrahler model and its application to simulate the organic and inorganic C balance of the Seine River for the period 2010-2013. The work is original and could be suitable for a journal like HESS. At its present state however, the manuscript is rather weak, in particular because of quite poor writing. I have also some concerns regarding model description and the evaluation and discussion of model results. Substantial revisions are required before I can recommend publication of this manuscript. Please, find my comments below.

Major comments:

**1: Writing**

The manuscript is poorly written. In particular the introduction and abstract are very weak, mainly because of bad English, but also with regard to structuring of the text and content-wise. It reads like someone wrote this in a great hurry with no time to read through the text again. I suggest that the authors put much effort into rewriting the manuscript. Results and discussion sections read fortunately better. Moreover, I would like to suggest that the authors try to get professional help for proofreading.

**#2: Alkalinity**

I am a bit confused by your use of total alkalinity (TA). To my understanding, TA is the sum of carbonate alkalinity (sum of charges of carbonate and bicarbonate ions) and non-carbonate alkalinity (incl. charges of ammonium, phosphate, silicate, borate and organic ions).

You state that you would need only two parameters to implicitly define all elements of the carbonate system, which is basically correct. But you say you would use DIC and TA for that. You could use DIC and carbonate alkalinity to calculate CO2 concentrations, for instance. But using TA instead would lead to erroneous result because of the non-carbonate contributions to TA. I see that you are representing ammonium and phosphate in your model, and it seems like they are included in TA in the model. But it is not clear to me whether you subtract ammonium and phosphate from TA to calculate carbonate alkalinity, and use that to calculate CO2. Here, I would like to see a much more detailed description of how you actually calculate CO2 concentrations and pH, including equations. Also I would like to see an equation that defines TA in your model, to see which ions are actually taken into account. Last but not least, I find it very strange that you report TA in  $\mu$ mol L-1, and not in  $\mu$ eq L-1 like it is normally done.

**#3: Water temperature**

It is not clear to me in how far the effects of water temperature on water viscosity (impact on k) and solubility of CO2 (impact on pCO2 and emission flux) are taken into account by the model. The seasonality in water temperature could have an effect on
the seasonality of CO2 concentrations, with a tendency for higher concentrations at lower temperatures. Here, I would like to see some clearer description in the method section, and maybe also some discussion in how far water temperatures could affect the weak performance of the model to reproduce the seasonality in CO2 concentrations, in particular in the higher stream orders (Fig. 6).

**4 Uncertainty sources in the model vs. observation based estimates**

When comparing simulations with observation based estimates, you should take into account more carefully that the uncertainties related to k or total river surface area can have a very different impacts.

When you use observed (or calculated) CO2 concentrations (or better partial pressures) to estimate the total CO2 evasion flux, you will first calculate the water-air CO2 gradient and multiply that by the estimates of k and the total stream surface area. That means that uncertainties related to the estimates of k and the total stream surface area will have a direct and proportional impact on the uncertainties related to the estimated total CO2 evasion flux. If you calculate the CO2 evasion rate per water surface area, only the uncertainty related to k matters, but not that of the total stream surface area.

When you use a process based model that represents the different sources of CO2 to the stream network, the choice of gas exchange velocity will have a substantial impact on simulated CO2 concentrations (as you have shown in Figure 7), but not on the CO2 emission flux (when talking about annual fluxes). For instance, Lauerwald et al. 2017 GMD found for their model on the Amazon River that increasing or decreasing k by 50% does not lead to a significant change in simulated aquatic CO2 emissions. This is because over a large river network, aquatic CO2 emissions will be close to the total of CO2 inputs (external inputs plus instream net-heterotrophy). If a too small k is chosen, CO2 will concentrate in the water column until a higher water-atmosphere CO2 gradient is reached that allows for a total river CO2 emission that is close to the sum of the CO2 inputs minus instream production (i.e. too high simulated CO2 concentration

HESSD
in the water column). Similarly, when a too high k is chosen, the total CO2 emission cannot exceed total CO2 inputs, and the too high k will be compensated by a too low water-atmosphere CO2 gradient (i.e. too low simulated CO2 concentration in the water column). In Figure 7, you have shown the impact of the choice of k on the simulated CO2 concentrations. I suggest that you also report the different CO2 emission fluxes that you simulated based on the different k-values. Based on that, you can maybe show that the choice of k does not have a too big impact on your IC balance calculation. But that leaves the impact of k on the CO2 concentration and pH. Could you maybe also show if and how the choice of k impacts the simulated pH?

When comparing to your earlier study to estimate the CO2 emissions from the Seine (e.g. L607-611), you should also discuss the estimate total river surface area as source of uncertainty. Similar to k, this uncertainty won't significantly affect your simulated total CO2 emission. However, when you calculate the CO2 emission rate per water surface area from your simulation results, the uncertainty related to river surface area estimate does have a direct and proportional effect on the uncertainty of emission rates. That means in this case, if the simulated CO2 emission rate per water surface area is too high, this is maybe because your estimate of the total stream surface area is too low!

General comments

**Abstract**

L16-38: The abstract needs better structuring. At the beginning in particular, after the first sentence, you should quickly explain the reasons of developing and applying a process-based model like you did. What are the specific research questions a model like this could help you with?

L20: Remove the commata around pyNuts-Riverstrahler

- L21: Replace "implemented on" by "applied to" .
- L23: By "diffuse constraints", do you mean "diffuse sources"? Please, clarify.
L24: Replace "characterised" by something like "assessed".

L25: Remove "In average,".

L26: WWTP has not been defined.

L18-27: Please, state over which period you have applied the model.

L33-38: The comparison to the 1990's comes out of nowhere. It's not clear why this comparison is made, what it implies, and where the data come from (are they also modelled in this study, or are they taken from another study?).

Introduction

L61-64: "as plant detritus, soil leaching or soil erosion and groundwater supply" This doesn't make sense. You are mixing characteristics of the carbon and sources of carbon in the same list. Better write something like "as plant detritus, organic carbon bound to eroded soil particles and organic acids which are brought in by runoff and drainage from soils".

L64: Delete "sources"!

L67-69: That doesn't make any sense.

L106-108: That should go to conclusion and outlook.

Materials and methods

L111: Degree signs needed.

L114-115: replace "annual water flow" by simply "water flow" because you report anyway the average flow over a longer period, and moreover, you report that flow in volume of water per second, and not per year.

L195-197: Please shortly list the characteristics which are represented.

L216-219: You state that you could use two variables, DIC and TA, to calculate all other
components of the carbonate system. Here I have to disagree. You could be doing this with DIC and carbonate alkalinity, but not with TA which is the sum of carbonate alkalinity and other sources of alkalinity including phosphate, silicate, ammonia and organic ions. But as you represent at least phosphate, ammonia and silicate, you can derive carbonate alkalinity from TA. Is that maybe what the model is doing? If yes, please clarify. But it would mean that you use more variables than just DIC and TA to calculate for instance CO2 concentrations.

L229: "CO2 gradient concentrations" should be "CO2 concentration gradients"

L265-269: How have these studies refined that approach? Did they simply re-calibrate the annual average concentration? Are these average concentrations adapted for different land use types, soil types, etc.? Or do you use only one average concentration per nutrient species which you apply everywhere? Please, clarify.

L278-281: Are these degradability classes defined somewhere? What is the basic turnover time or decomposition rate for each class under some sort of standard condition (which needs to be defined)?

L291-292: Here you should clarify if these degradability classes have the same turnover rates as those for DOC, or if they are defined differently. Otherwise, this statement might be confusing.

L293-297: Do you really mean TA here? Or maybe carbonate alkalinity? Note that phosphate, ammonia, silicate and organic ions count into TA.

L299: You should write mg CO2-C L-1 instead of mg C-CO2 L-1. Figure 3: How can alkalinity be reported in  $\mu$ mol L-1? Do you mean  $\mu$ eq L-1? Also, you should report DIC in  $\mu$ mol L-1 to be consistent, even if you report alkalinity in  $\mu$ eq L-1 (which you definitely should!).

L339-340: Could you please give the implied average concentration of free dissolved CO2 for these effluents?

HESSD
L420: What do you mean by "good levels"?

phosphate and ammonia contributions to TA?

Results

Figure 6: For the river network within and upstream of Paris, the model shows a very weak performance with regard to seasonality in CO2 concentrations and pH. There appears to be a systematic underestimation of TA throughout time and space. That would have to be discussed. Moreover, I wonder if a simple recalibration could help to simply solve this problem.

L368-370: Did you have additional hydrochemical data available to correct for at least

L459-462: Raymond et al. 2012 trained their empirical model for k on relatively small rivers (defined by discharge). As you have discussed before, the equations by Alin et al. may only be valid for a stream width up to 100 m. Also Raymond et al.'s equation is only valid up to a certain discharge. Following that same logic, you cannot apply their equations here. These issues should be discussed here.

L463-466: As discussed in Alin et al., in wider rivers, wind stress might become the dominant control of k. It seems to be potentially problematic to just omit the term related to wind speed in the equation by Ho et al.. I would expect that the underestimation of k might arise from that. You should quantify that potential bias for a realistic range of wind speed, and discuss why you think that this bias would be negligible. Wouldn't it be better to simply assume an average wind speed? Or you could simply use average monthly wind speed values per stream order from e.g. http://worldclim.org/version2.

L469: Here section 4.1 follows after section 3.1.4.. I have the strong feeling that some sections have gone missing here. But I hope it's just some stupid mistake with numbering. I will simply assume that this is still the results section, and discussion section starts in L550.

L492-493: I assume that "ventilation" means CO2 emissions from water surface. Any-

HESSD
way, you should use a more consistent terminology to not confuse the reader.

Discussion

L554-555: Öquist et al. found that for which river? In how far is that river comparable with the Seine river?

L562-569: You could still calculate the average wind speed per stream order and simply use that in your equation. Also, you could simply adapt k empirically in a way that optimizes the fit between observed and modelled CO2 concentrations.

L589-594: Temporal dynamics in CO2 are likely the strongest control on the temporal dynamics of pH. As long as you don't get those right, you won't be able to reproduce pH, no matter what formulation you will use.

L610-618: Here you should mention how much SO1 contributes to the total CO2 emission and to the total stream surface area of the Seine river network. Then you could give the average CO2 emission rate per stream surface area for SO2-SO7 only. Like this, you could support your statement with numbers.

L684-686: Your results do not support this conclusion. I particular the performance with regard to reproducing observed CO2 concentrations is quite bad, and a decent discussion on why that is missing so far.

HESSD

---

## Referee Comment (RC3) · Anonymous Referee #3 · 29 Jan 2020

**Modeling inorganic carbon dynamics in the Seine River continuum in France by Marescaux et al.**

The authors present a modeling effort of inorganic carbon dynamics in the Seine River. It is done in the pyNuts-Riverstrahler model. With the new module, the outgassing of $CO_2$ is calculated for the time period 2010-2013. Also a budget for inorganic and organic carbon including alkalinity for the whole Seine river basin is presented. The manuscript is well structured. The model performance from small orders to higher orders is reasonable at first sight. However, considering how sensitive the balance between alkalinity – $CO_2$ – pH is, the model performance from small orders to higher orders is impressive. I recommend to publish this paper after major revision.

**Specific comments**

- There are many well tested and well described inorganic carbon modules readily available (see for a review: Orr et al., 2015, https://www.biogeosciences.net/12/1483/2015/bg-12-1483-2015.pdf). Is there a specific reason to develop an own implementation for pyNuts-riverstrahler?
- In paragraph a kind of sensitivity analyses is presented for the gas transfer velocity. It is not clear to me, why this parameter is chosen. I miss a more extended model sensitivity analyses to determine which input parameters are sensitive to $CO_2$ emissions or carbon export to the sea. Which model parameter contributes most to variability of $CO_2$ emissions?

**Technical corrections main text**

1. Double equation numbers. Equation numbers in SI and in main article overlap. Please give them different names.
2. Line 46: The first highlight of a successful implementation. I was surprised by this highlight. There is no word on the implementation details in this article. I think the model itself is never a highlight. The model is a tool to show some of your findings (as you do in this article). So remove.
3. Line 101: Again purpose of this study is an implementation. I don't think this journal is suited for this purpose.
4. Line 102: "pyNuts modeling environment" I would like to have a reference to this. To me it is not clear what the difference is between RiverStrahler, RIVE pyNuts-Riverstrahler. All names are used here. Please elaborate this.
5. Line 106: remove s from works
6. Line 111: Add unit to the decimal numbers.
7. Lines 147 – 154: This footnote is unclear. Last line: calculation of stream velocity. How? Is something fallen of the page here? Use of parameter WSA is confusing. It could mean: mean_width * Slope * Area (not defined here). Change name or put bracket around name.
8. Line 161: Please make figure captions consistent. Figures 1,5,6,.. ends with a dot, but other figure captions not.
9. Lines 192-197: Message in this paragraph is unclear

10. Line 210: Which module? I only see RIVE in figure 2, including TA and DIC. Highlight the IC module in figure 2.
11. Line 236: Eq 3 is referred to as eq 1 in SI
12. Line 238: Table 2: It is not clear how column TA is made out of the formulas 3 – 8. Please explain.
13. Line 258: values and constants are given in Table 2. Is this reference correct? I don't see them.
14. Line 263: Where are the subsurface and groundwater flow components described? Is this in line 201 and further?
15. Line 296: Are pH values measured? From $HCO_3^-$ and pH, the $CO_2$ concentrations could be calculated.
16. Line 318: S3. Is this the right reference?
17. Line 343-351: Reservoirs are an integrated component of the river network itself. They are not point sources, they are receivers of alkalinity. This is a strange paragraph. There is no module with DIC module for reservoirs, so measurements from one reservoir are taken. Does this mean that reservoirs are not part of the module? This can't be true….
18. Line 400: Figure 4: missing x-axes like for example "Strahler order".
19. Line 402: Change mgC- $CO_2$ $L^{-1}$ to mgC $L^{-1}$
20. Line 409: werefollowed in were followed.
21. Line 438: Figure 6 is too small to see the results.
22. Line 439: Subscript of CO2 (twice)
23. Line 440: What is simulation envelope? Can I see this? What is the gray area?
24. Line 448: Here a time lag is mentioned. But size is total different as well. I don't see any explanation for this.
25. Line 451: There is a four (number with dot) shown. Delete.
26. Line 461: to = too
27. Line 522: Subscript of CO2
28. Line 545: Figure 9, to show the spatial dynamics of the ecology in the continuum, it might be interesting to explicitly present the relative contribution of benthic primary producers and the planktonic primary producers to the total primary production.
29. Line 563: Did you test the performance of the model with the wind speed parameterization suggested by Alin et al. 2011?
30. Line 584: Any sense of direction which specific algae parameter(s) / trophic condition(s) has/have changed that causes the temporal variability not matching?
31. Line 594: Dot at end of line.
32. Line 602-604: What is the contribution of estimated k-value to the uncertainty of the total basin $CO_2$ emissions? You slightly touch upon in figure 7, but basin total $CO_2$ emissions are not mentioned.
33. Line 604-606: I would not compare outgassing by surface area to global studies. Reference to temperate rivers are relevant.
34. Lines 613-614: Sentence is not correct.

35. Line 620-624: The OC export estimate by Meybeck is higher, but the detail and scale of his study is incomparable to yours. How do you know erosion in the Seine is limiting for OC export compared other temperate rivers? Also, what makes the trophic state of the Seine other than other temperate rivers?
36. Line 622: change ": )".
37. Line 624: Add ( before Rocher.
38. Line 646: I would add benthic information to figure 9 too
39. Line 660-661: I don't see benthic respiration explicitly mention in figure 9.
40. Line 668: Figure8 add blank.
41. Line 693-694: Where do you show small orders are driven by groundwater discharges?

**Technical corrections SI**

42. Page 1 and 8 : broken link.
43. Page 2 : *** Now it is added to model RIVE ??
44. Figure S1 : I see nine red hatching areas. Not eight. Please change this also in main text (if 9 is the correct number).
45. Eq 6 does not make sense here. Remove. Will be given in eq 14 and 15.
46. Eq 11 : Remove C from $K_2C$
47. Section 3: Eq 17 to 19: What is CA? Carbonic Acid? Carbonate Alkalinity?
48. Eq. 28 should be $k600 = 13.82 + 0.35v$
49. Reference list: I would like to have one for the SI and one for the main text. Please also check the reference list. I was looking for Milero et al. 2006. It is used in the text (Table S1), but not mentioned in reference list.

---

## Author Comment (AC1) · 11 Feb 2020

**Response to anonymous Referee #1**

This paper described a biogeochemical model incorporating inorganic carbon cycle and applied the model to the Seine River system. The model was built based on an existing biogeochemical model and the model structure and setup have been sufficiently described. The results from current study help to fill up the gaps in understanding the contribution of inland waters to the global carbon cycle. However, the model performance is not very convincing. There had been a few other models able to simulate inorganic carbon in rivers and have not been discussed. In summary, the manuscript has potential to be improved and I would like to suggest the authors to consider:

1. improving the model performance presentation (see specific comments below);

2. the discussion of current findings is too site-specific; I would suggest to expand the discussion to a more general sense, e.g. how the inorganic carbon system in Seine compared to other inland water systems? What are the meaning of current findings to estimating the roles of rivers in local and global carbon cycle? etc.

3. Also, the text writing in the introduction and discussion need to be polished. I list a few issues in the specific comments below but encourage the authors to go through the text and improve the writing in general.

We thank the reviewer for his comments and advice on how to improve the manuscript, especially the model performance presentation, where the text has significantly evolved.

We now discuss more generally the merits of a modelling approach in comparison with other measurement based CO2 emission estimates. Also, we have tried to replace our finding for the Seine River system to a broader context of aquatic CO2 evasion from temperate and/or human impacted river systems, providing comparative values.

Although the manuscript has been already revised by a professional English native person, we submitted the revised manuscript for another complete proofreading in order to improve the English writing.

Specific comments:

Line 37-38: some words are missing from this sentence. 'Outgassing was the most important {carbon sink/inorganic carbon process}?

**A1.** We modify the sentence as: "*The most significant outgassing was in lower order streams while peaks were simulated downstream of the major wastewater treatment effluent.*" [L32-34]

'Line 69-71: This statement seems controversial to some other findings that eutrophic system usually contains richer organic matters and pCO2 (e.g. Borges and Abril, carbon dioxide and methane dynamics in estuaries, DOI: 10.1016/B978-0-12-374711-2.00504-0). Can you please explain more about this statement?

**A2.** Thank you for your comment. Indeed, we wanted to highlight that some ecosystems can be a source and other a sink of $CO_2$. We now modify and precise that the statement is for lentic eutrophic systems and we change 'can be' by 'may be'.

*"As a whole, oligo- and mesotrophic lotic hydrosystems generally act as a source of carbon while surface water of lentic eutrophic systems may be undersaturated with respect to atmospheric $pCO_2$ (Prairie and Cole, 2009; Xu et al., 2019; Yang at al., 2019)." [L68-71]*

Line 69-71: The Xu et al. 2019 reference is missing;

**A3.** Thanks, we added the references:

- Xu, Y. J., Xu, Z. and Yang, R.: Rapid daily change in surface water $pCO_2$ and $CO_2$ evasion: A case study in a subtropical eutrophic lake in Southern USA, J. Hydrol., doi:10.1016/j.jhydrol.2019.01.016, 2019.
  https://www.sciencedirect.com/science/article/pii/S0022169419300599?via%3Dihub
- Yang, R., Xu, Z., Liu, S. and Xu, Y. J.: Daily pCO2 and CO2 flux variations in a subtropical mesotrophic shallow lake, Water Res., doi:10.1016/j.watres.2019.01.012, 2019.
  https://www.sciencedirect.com/science/article/abs/pii/S0043135419300466?via%3Dihub

Line 72-76: This statement needs to be treated carefully. Other methods, such isotope surveys, can also be used to investigate the fate of carbon in aquatic systems.

**A4.** Thanks, we agree with your comment and modify the sentence as:

*"Direct measurements of $pCO_2$ or isotopic surveys (as realized by Dubois et al. 2010 in the Mississippi River) along the drainage network are still too scarce to accurately support temporal and spatial analyses of $CO_2$ variability. While calculations from pH, temperature and alkalinity may help reconstruct spatiotemporal patterns of $CO_2$ dynamics (Marescaux et al., 2018b), modeling tools can predict the fate of carbon in whole aquatic systems." [L72-76]*

Line 85-90: A few early papers had reported models including the inorganic carbon cycle and pCO2 exchange but have not been mentioned here. Such as the CONTRASTE model (Vanderborght et al 2002, Application of a transport-reaction model to the estimation of biogas fluxes in the Scheldt estuary, Biogeochemistry 59: 207-237), RTM model (Regnier et

al 2013, modelling estuarine biogeochemical dynamics: from the local to the global scale, Aquat Geochem 19: 591-626); How is the current model compared to these models?

A5. The CONTRASTE and the RTM models are estuarine models and we initially refer only to river models, but we added now these two references .
However, the main differences between the formalisms of pyNuts-Riverstrahler (a river model) and these estuarine models lie in the description of the phytoplankton groups, organic carbon matter and benthic activities which are more detailed in pyNuts-Riverstrahler, while these estuarine models described the shape of the estuary and take into account the tides, the salinity and the wind.

Estuaries are highly reactive systems from a biogeochemical point of view, also with proportionally greater gas exchanges at the water-atmosphere interface because of the river section enlargement in these area. In the case of the Seine, it is worth to mention to the reviewer that we recently carried out an integrated modelling approaches, by coupling the Riverstrahler model to the C-GEM estuarine model (developed by the same team of the RTM and CONTRASTE models), which made it possible to specify the respective ecological functioning and contributions of the fluvial and estuarine parts in the organic and inorganic carbon budgets.

Laruelle, G. G., Marescaux, A., Gendre, R. Le, Garnier, J., Rabouille, C. and Thieu, V., Carbon dynamics along the Seine River network: Insight from a coupled estuarine/river modeling approach, Front. Mar. Sci., doi:10.3389/fmars.2019.00216, 2019

Line 91-92: This individual sentence as one paragraph is not reading well. Can be merged with next paragraphs.

A6. Thanks, we merged the two sentences as:

*"The Seine River (northwestern France) has long been studied using the biogeochemical riverine Riverstrahler model (Billen et al., 1994; Garnier et al., 1995), a generic model of water quality and biogeochemical functioning of large river systems." [L92-94]*

Line 111: unit of the north and east coordinates?
A7. Thanks we added the coordinates:

Situated in northwestern France within (decimal degrees) 46.95° –50.01° north and 0.11° – 4.00° east.

Line 228-230: the gas transfer velocity only affect the exchange rate, not the change direction of pCO2 (and therefore DIC).

A8. We changed the sentence to make things clearer:

*"The exchange of $CO_2$ between the water surface and the atmosphere depends, respectively, on the gas transfer velocity (k-value) and on the sign of the $CO_2$ concentration gradient at the water surface–atmosphere interface (S3.5). Change in $pCO_2$ will in turn affect DIC concentrations (see Table 2, Eq. 1)."* [L231-234]"

Line 383-384: why only 4 years simulated but NRMSE were performed on inter-annual variations per decade, instead of 2010-2013? Also, normalized against mean observational data instead of inter-annual variations is more representative.

**A9.** We performed NRMSE analysis on inter-annual variations per decade because the aim was to also evaluate the ability of the model to represent the seasonal trends. Because of the small amount of observations available for each year and for each 10-days period (especially for DIC concentrations), we preferred to average the available inter-annual values per 10-days period (which is actually the resolution of the RIVERSTRAHLER model). We choose to normalize the RMSE by the inter-annual variation because the mean of observations are not representative of the observations that can take extreme values.

Line 402-404: as CO2 concentrations are related to DIC and TA, it would be better if you show the comparisons of observed and modelled DIC/TA along with the CO2 concentrations.

**A10.** We fully agree. We do not have enough observation data especially in the upstream part of the Seine drainage network to propose similar analysis for DIC and TA by stream order. However, at the section "1.3.Seasonal variations", we selected 4 stations with enough data available in an upstream-downstream gradient to jointly analyze the variations of observed CO2, TA and DIC and compare them to the model.

Line 517-518: can't find CO2 outgassing in figure 9?

**A11.** We corrected the typo : 'figure 8'

Figure 6: why there are two dark lines in the water flow of the outlet of the basin? Also, as the model timeframe includes dry and wet years, it is better to show the results year to year but not averaged from 4 simulated years;

**A12.** there is an error in the plot. One of the two black lines is in fact the link between the average observation points and should not have to be drawn.

Because of the lack of observation data (especially for DIC and CO2), we decided to provide average values and to assess the model performance using simulation averaged on this 4-years timeframe.

Also, looking at the standard deviations of observed discharge values, it could be seen that hydrological regimes were not so different over the 2010-2013 timeframe (e.g. drier in summer 2011). This is mostly explained by the water regulation by reservoirs occurring in

the upstream part of the river basin. Impact of this flow regulation is evident upstream of Paris, then fades downstream and this is clearly visible when looking at the increase of observed discharges standard deviations from upstream Paris to Poses

Line 583-589: this sentence needs to be re-organized.

**A13.** The whole paragraph has been reorganised:

*"Also, despite the fact that the biomass level of phytoplankton was consistent with the observations, the seasonal pattern was not satisfactory reproduced by the model. However, it is worth mentioning that phytoplankton parameters in RIVE were determined through laboratory experiments at a time when the amplitude of algal blooms was much higher than at present (up to 4.5-6 mgC $L^{-1}$ i.e., chlorophyll a reaching 150 µgChla $L^{-1}$, Garnier et al., 1995). Indeed, the implementation of the European Water Framework Directive in the 2000s with enhancement of treatments in WWTPs greatly improved water quality (Romero et al., 2016). New laboratory experiments for possibly taking into account additional phytoplankton groups or species in these new trophic conditions and/or mixing stochastic and mechanistic modeling are required to better represent phytoplankton temporal dynamics in the model. In addition, the observed incident light, instead of the empirical relationship used, would improve the early winter bloom, newly occurring in a changing environment"* [L611-622].

Line 624: left bracket is missing in citation;
**A14.** Thanks, we added it.

Section 4.3: is there a relationship between the river eutrophic state and the metabolism activity, and CO2 outgassing?

**A15.** Eutrophic state of the river indeed changes the metabolism activity (see Garnier & Billen, 2007). We observe that the influence of the metabolism activities on the $CO_2$ outgassing is low. Indeed, in the carbonated Seine River, the IC originating from groundwater supports the $CO_2$ outgassing along the network (figure 8). Nevertheless, instream metabolism activities produce or consume $CO_2$. In high stream Strahler orders, river metabolism activities (as NPP and heterotrophic respiration) influence seasonal variations of $CO_2$ concentrations (see figures below).

[Figure]

NB: SO7 with a scale change for $CO_2$:

[Figure]

We added this remark in the manuscript:

*"We observe that the influence of the metabolism activities on the $CO_2$ outgassing is low. Indeed, in the carbonated Seine River, the IC originating from groundwater supports the $CO_2$ outgassing along the network (Figure 8). Nevertheless, instream metabolism activities produce or consume $CO_2$."* [L684-687]

---

## Author Comment (AC2) · 11 Feb 2020

The manuscript submitted by Marescaux et al. presents a technical upgrade of the pyNuts-Riverstrahler model and its application to simulate the organic and inorganic C balance of the Seine River for the period 2010-2013. The work is original and could be suitable for a journal like HESS. At its present state however, the manuscript is rather weak, in particular because of quite poor writing. I have also some concerns regarding model description and the evaluation and discussion of model results. Substantial revisions are required before I can recommend publication of this manuscript. Please,find my comments below.

**A1.** We thank the reviewer for his/her comments and advice on how to improve the manuscript. We have taken into account all of his advice. The manuscript has been already revised by a professional English native person. However another complete proofreading of the article has been done again after our own revision to make the paper stronger and in order to improve the English writing.

Major comments:

**1: Writing**

The manuscript is poorly written. In particular the introduction and abstract are very weak, mainly because of bad English, but also with regard to structuring of the text and content-wise. It reads like someone wrote this in a great hurry with no time to read through the text again. I suggest that the authors put much effort into rewriting the manuscript. Results and discussion sections read fortunately better. Moreover, I would like to suggest that the authors try to get professional help for proofreading.

**A2.** We restructured the text and following your advice we sent the manuscript for professional proofreading.

**2: Alkalinity**

I am a bit confused by your use of total alkalinity (TA). To my understanding, TA is the sum of carbonate alkalinity (sum of charges of carbonate and bicarbonate ions) and non-carbonate alkalinity (incl. charges of ammonium, phosphate, silicate, borate and organic ions).

You state that you would need only two parameters to implicitly define all elements of the carbonate system, which is basically correct. But you say you would use DIC and TA for that. You could use DIC and carbonate alkalinity to calculate $CO_2$ concentrations, for instance. But using TA instead would lead to erroneous result because of the non-carbonate contributions to TA. I see that you are representing ammonium and phosphate in your model, and it seems like they are included in TA in the model. But It is not clear to me whether you

subtract ammonium and phosphate from TA to calculate carbonate alkalinity, and use that to calculate CO2. Here, I would like to see a much more detailed description of how you actually calculate CO2 concentrations and pH, including equations. Also I would like to see an equation that defines TA in yourmodel, to see which ions are actually taken into account. Last but not least, I find it very strange that you report TA inµmol L-1, and not inµeq L-1 like it is normally done.

**A3.** We agree with the reviewer that total alkalinity TA is could be defined as:

$$TA \equiv 2[CO_3^{2-}] + [HCO_3^-] + [H_2BO_3^-] + 2[HBO_3^{-2}] + 3[BO_3^{-3}] + [OH^-]$$
$$+ [organic/inorganic\ H^+acceptors] - [H^+]$$

In our approach TA is defined by terrestrial boundary conditions (point and diffuse sources, see TA inputs Eq. 9). TA concentrations were measured in ground waters and in headwater streams. TA is then affected along the simulations by heterotrophic planktonic respiration of bacteria, zooplankton and benthic bacteria, nitrification, denitrification and photosynthesis (see Eq. 10) according to the stoichiometry defined in table 2.

TA and DIC are used to calculate the pH as proposed by Culberson (1980). The equations of Culberson were derived with the assumption that only bicarbonates, carbonates and borates contribute to TA. The author specifies that phosphate concentration $< 3.10^{-6}$ mol/l and silicate at concentrations $< 50.10^{-6}$ mol/l have negligible effect on the calculation of the pH ($< \sim0.001$ pH). In addition, total dissolved boron concentration can generally be ignored in freshwaters (Emiroglu et al., 2010).

So in the carbonated freshwaters of the Seine River we make the assumption that for the pH calculation TA can be used as an approximation of CA. We added this remark in the supplementary material section S3.

Regarding the detailed equations for pH calculation, there are provided in the supplementary information "3.4"

Nevertheless, in this later section, we wrongly refer to carbonate alkalinity (CA) instead of Total Alkalinity (TA). This error probably misled the reviewer, making him/her think that we were recalculating the carbonate alkalinity based on the total alkalinity and ammonium + phosphate ions (which is not the case, we only use TA in our approach, as simplified for freshwater, see above).

Regarding the reviewer remark about units used for alkalinity. In biogeochemistry modeling, total alkalinity used to be described in meq/L however more and more manuscripts described it now in µmol/L since chemical formula enable to make the conversion (among others: Borges, A. V. and Abril, G.: Carbon Dioxide and Methane Dynamics in Estuaries., 2011.; Regnier, P., Arndt, S., Goossens, N., Volta, C., Laruelle, G. G., Lauerwald, R. and Hartmann, J.: Modelling Estuarine Biogeochemical Dynamics: From the Local to the Global Scale, Aquat. Geochemistry, doi:10.1007/s10498-013-9218-3, 2013).

We decided to keep this unit.

**3: Water temperature**

It is not clear to me in how far the effects of water temperature on water viscosity (impact on k) and solubility of $CO_2$ (impact on $pCO_2$ and emission flux) are taken into account by the model. The seasonality in water temperature could have an effect on the seasonality of $CO_2$ concentrations, with a tendency for higher concentrations at lower temperatures. Here, I would like to see some clearer description in the method section, and maybe also some discussion in how far water temperatures could affect the weak performance of the model to reproduce the seasonality in $CO_2$ concentrations, in particular in the higher stream orders (Fig. 6).

**A4.** At this stage the Riverstrahler model does not include a proper thermic model. A mean temperature function (reproducing seasonal variations) is provided for each stream order as boundary condition, as described in Billen et al 1994. We adjusted the parameters of this empirical temperature function for each Strahler order according to measurement available for the recent period. Results of this calibration for observed water temperature averaged by 10 decade over time period 2006-2016 is provided here after:

[Figure]

We can observe that the equations used enable a good representation of averaged water temperature variation for each Strahler order. Then, the weak performance of the model to reproduce the seasonality in $CO_2$ concentrations cannot be explained by the water temperature.

We added a sentence in the methodology :

*"Water temperature was calculated according to an empirical relationship, adjusted on inter-annual averaged observations (2006—2016), and describes seasonal variation of water temperature in each Strahler order with a 10-days time step (see S2)." [L200-202]*

The section 3.5 in SM3 describes in detail how temperature is taken into account to calculate *k*-value (Eq. 26 and 27).

Solubility is calculated according to Weiss (1974) and the reference is provided in section S3.6 table 1. We added a reference to this table in the manuscript:*"The different values of constants and parameters used in the inorganic carbon module are introduced in Table 1 of S3.6. The full inorganic carbon module is described in S3 (3.1 to 3.5)." [L262-264]*

We also modified the discussion section to better explain the possible factor limiting the performance of our model in the representation of $CO_2$ seasonality (temperature, hydrology, phytoplanktonic biomass etc.):
*"The model showed a weak performance in representing $CO_2$ seasonality. Referring to a previous study (Marescaux et al., 2018b), $pCO_2$ seasonality in the Seine River resulted from a combination of water temperature and hydrology leading to an increase in $pCO_2$ and $CO_2$ evasion fluxes from winter to summer/autumn. The pyNuts-Riverstrahler model however has an accurate representation of these constraints and would not account for these discrepancies. Also, despite the fact that the biomass level of phytoplankton was consistent with the observations, the seasonal pattern was not satisfactory reproduced by the model. "*
*[L606-612]*

**4 Uncertainty sources in the model vs. observation based estimates**

When comparing simulations with observation based estimates, you should take into account more carefully that the uncertainties related to *k* or total river surface area can have a very different impacts.

When you use observed (or calculated) CO2 concentrations (or better partial pressures) to estimate the total CO2 evasion flux, you will first calculate the water-air CO2 gradient and multiply that by the estimates of k and the total stream surface area. That Means that uncertainties related to the estimates of k and the total stream surface area will have a direct and proportional impact on the uncertainties related to the estimated total CO2 evasion flux. If you calculate the CO2 evasion rate per water surface area,only the uncertainty related to k matters, but not that of the total stream surface area.

When you use a process based model that represents the different sources of CO2 to the stream network, the choice of gas exchange velocity will have a substantial impact on simulated CO2 concentrations (as you have shown in Figure 7), but not on the CO2 emission flux (when talking about annual fluxes). For instance, Lauerwald et al. 2017 GMD found for their model on the Amazon River that increasing or decreasing k by 50% does not lead to a

significant change in simulated aquatic CO2 emissions.This is because over a large river network, aquatic CO2 emissions will be close to the total of CO2 inputs (external inputs plus instream net-heterotrophy). If a too small k is chosen, CO2 will concentrate in the water column until a higher water-atmosphere CO2 gradient is reached that allows for a total river CO2 emission that is close to the sum of the CO2 inputs minus instream production (i.e. too high simulated CO2 concentration in the water column). Similarly, when a too high k is chosen, the total CO2 emissions cannot exceed total CO2 inputs, and the too high k will be compensated by a too low water-atmosphere CO2 gradient (i.e. too low simulated CO2 concentration in the water column). In Figure 7, you have shown the impact of the choice of k on the simulated CO2 concentrations. I suggest that you also report the different CO2 emission fluxes that you simulated based on the different k-values. Based on that, you can maybe show that the choice of k does not have a too big impact on your IC balance calculation. But That leaves the impact of k on the CO2 concentration and pH. Could you maybe also show if and how the choice of k impacts the simulated pH?

**A5.** It seems important to us to repeat here that the *k*-values modification tests only concern the downstream parts of the network (order 6 or 7 greater than 100), i.e. a total of 367 km out of the 24,306 km of the Seine network.

We understand the reviewer's suggestion on the impact of different single *k*- values applied to an entire hydrosystem on IC balances. However, this work does not primarily aims at working on the sensitivity of *k*- values. We have chosen the formulation of Alin (2011) applicable to the great majority of the Seine network, and we only propose a second formulation for the last hundred kilometres to better take into account a specific feature of the basin Seine (the huge Seine Aval wastewater treatment plant).

The presence of a large wastewater treatment discharge (6 million Inahb. Eq) makes $CO_2$ concentrations very sensitive to formulation of *k*- value in this downstream sector (as shown in the figure 7). Such an impact on $CO_2$ concentrations, directly affects pH, showing abrupt decrease when $CO_2$ concentrations increase right after the WWTP release, and then an increase concomitant with the decreasing of $CO_2$ concentrations.

[Figure]

Nevertheless, we followed the reviewer's remarks and estimated the impact of these *k*-values variations in the most downstream parts (width >100m) on the total emission of the Seine network. For the 4 formulations tested (wide > 100m; formulations tested only on 1.5% of the total length of the drainage network), the variations in the IC balance is up to 6.18%. Consequently, we have modified the manuscript in the following way:

- We better explain the test performed on the k formulation (restricted to order 6 and 7):
    *"Different values of k were explored specifically in the downstream part of the Seine river network (SO6 and SO7 where river width exceeds 100m) (Figure 7)"* [L464-466]

- better discuss the impact of changes in k with respect to the IC balance with reference to the work of Lauerwald et al. 2017

Thanks to the suggestion of the reviewer, we were interested in comparing our work with that of Lauerwald et al. (2017). As described by the reviewer, Lauerwald et al. (2017) found for their Amazon River model that a 50% increase or decrease in *k*-value does not result in a significant change in simulated aquatic $CO_2$ emissions.

New simulations were performed in order to compare the $CO_2$ emission estimates using different *k*-formulations. In addition to the simulation selected in our manuscript (here call k_Reference), we calculate emissions when the *k*-values were formulated as:

   - Alin et al. (2011) (equation <100 m) (k_Alin) all along the drainage network

   - Raymond et al. (2012) (Table 5 Eq. 2) (k_Raymond) all along the drainage network.

The results are presented in the table below. We also add $CO_2$ emissions estimated by Marescaux et al (2018a) based on observations.

| | Names | k-value SOs 1-6 (width < 100 m) | k-value SOs 6-7 (width > 100 m*) | $CO_2$ emissions (GgC yr-1) | Time period | Surface area of rivers (km²) |
|---|---|---|---|---|---|---|
| simulations | k_Reference | Alin et al. 2011 (equation < 100 m) | k mix of Ho et al. and O'Connor et al. without the wind term | 364 | 2010-2013 | 260 |
| | k_Alin | Alin et al. 2011 (equation < 100 m) | Alin et al. 2011 (equation < 100 m) | 388 | 2010-2013 | 260 |
| | k_Raymond | Raymond et al. 2012 (Table 2 equation 5) | Raymond et al. 2012 (Table 2 equation 5) | 418 | 2010-2013 | 260 |
| Estimations based on observations | Marescaux_2018 | Raymond et al. 2012 (Table 2 equation 5) | Raymond et al. 2012 (Table 2 equation 5) | 590 | 2010-2016 | 265 |

\* SOs 6-7 > 100m represent 367 km out of the 24,306 km of the river network until its outlet at Poses (either 1.5 % only)

**Comparison of the k_Reference and k_Alin simulations:** A change in the k-value on rivers with a width > 100m (representing only 1.5% of the total length of the Seine River) led to a difference in $CO_2$ emissions of 28 GgC yr-1 (6.18%). Alin et al. (2011) (<100m) equation cannot be used on wide rivers and the formulation using Ho et al. (2016) and O'Connor and Dobbins (1958) allows a better description of the longitudinal profile of $CO_2$ concentrations along the Seine.

**Comparison of the k_Reference, k_Alin and k_Raymond simulations:** Our estimates of $CO_2$ emissions do not confirm the statement of Lauerwald et al. (2017) that large variations of k (+/- 50%) lead to a marginal change in simulated aquatic $CO_2$ emissions (around 4%). Indeed, compared with the k_Reference, the simulations according to k_Alin increase $CO_2$ emissions from the river system by 5.6% and the simulations according to k_Raymond et al. 2012 increase $CO_2$ emissions by 15%.

A main difference with the work of Lauerwald et al. (2017) is that we used a more accurate *k* calculated at each time step (10 days) and at every kilometer of the river network (according to water temp., velocity, depth). In addition, Lauerwald et al. (2017) carried out simulations on a natural network without the huge organic carbon load brought by wastewater treatment plants in an urbanized system that disrupts carbon dynamics, like the SAV-WWTP (10 million Inhab. Eq) in the downstream part of the Seine river.

**Comparison of the simulations vs. Marescaux et al (2018)**

Our estimates of simulated CO2 outgassing are lower than our previous estimate based on observation (Marescaux et al. 2018a). This difference is explained below:

- Marescaux et al (2018a) use k formulates according to Raymond et al. (2012) all along the seine drainage network (not adapted for large river section) and $CO_2$ emission value is most likely overestimated

- Comparison between the k_Reference, k_Alin and k_Raymond simulations demonstrated that $CO_2$ emissions from the Seine are sensitive to k-formulation (until 15% difference).

- Among the 3 simulations we have compared (k_Reference, k_Alin and k_Raymond), only the k_Reference simulation takes into account a k formulation adapted for large river sections.

For these reasons, we believe that our estimate of 364 ± 99 GgC/yr, using a process based model, is a more accurate value of $CO_2$ emission from the Seine River. We also acknowledge that this value might be slightly underestimated with respect to Figure 4 (of the present paper) which shows that our simulated $CO_2$ concentrations were overestimated for SO1 but underestimated for SO2 to SO7.

We reformulated the following section in "4.1. Evaluation of the model" :

*"Future work with direct k measurements and/or a new representation of k-values in the model could help improve outgassing simulations with pyNuts-Riverstrahler. A test of different k formulations on high stream orders (width > 100 m) representing only 1.5% of the length of the river system showed an increase of the total $CO_2$ outgassing estimates by up to 6.2%. Our model is k sensitive and our estimates differs from the results of Lauerwald et al (2017), who observed that a large variation in k does not lead to a significant change in simulated aquatic $CO_2$ emissions. For the Seine River here, we indeed used a more accurate k-value calculated at each time step (10 days) and at every kilometers of the river network (according to water temperature, velocity, depth). In addition, a huge organic carbon load is brought by WWTPs in this Seine urbanized hydrosystem that disrupts carbon dynamics (e.g., WWTPs treating 12 million inhab. eq in the Parisian conurbation) in the downstream part of the Seine River, in contrast to simulations on a natural network (Lauerwald et al., 2017)."* [L579-590]

General comments

Abstract

L16-38: The abstract needs better structuring. At the beginning in particular, after the first sentence, you should quickly explain the reasons of developing and applying a process-based model like you did. What are the specific research questions a model like this could help you with?

**A7.** We have reformulated the abstract

L20: Remove the commas around pyNuts-Riverstrahler

**A8.** Commas have been removed.

L21: Replace "implemented on" by "applied to" .

**A9.** We replaced implemented by "developed" as this version take into account a new $CO_2$ module.

L23: By "diffuse constraints", do you mean "diffuse sources"? Please, clarify.

**A10.** Thanks we replaced diffuse constraints by diffuse sources

L24: Replace "characterised" by something like "assessed".

**A11.** Done

L25: Remove "In average,".

**A12.** Done

L26: WWTP has not been defined.

**A13.** Done

L18-27: Please, state over which period you have applied the model.

**A14.** The period is clearly stated in the right paragraph as:

*"During the period studied (2010–2013) ..." [L28]*

L33-38: The comparison to the 1990's comes out of nowhere. It's not clear why this comparison is made, what it implies, and where the data come from (are they also modelled in this study, or are they taken from another study?).

**A15.** We removed the mention to the 1990's, which is not necessary here. This refers to previous studies, mentioned in the discussion.

introduction

L61-64: "as plant detritus, soil leaching or soil erosion and groundwater supply" This Doesn't make sense. You are mixing characteristics of the carbon and sources of carbon in the same list. Better write something like "as plant detritus, organic carbon bound to eroded soil particles and organic acids which are brought in by runoff and drainage from soils".

**A16.** Thank you for this remark. We used your own sentence.

*"Organic carbon entering rivers can originate from terrestrial ecosystems as plant detritus, soil leaching or soil erosion and groundwater supply, but it can also be produced instream by photosynthesis or brought by dust particles (Prairie and Cole, 2009; Drake et al., 2017)"* [L61-64]

L64: Delete "sources"!

**A17.** We suppressed "sources"

L67-69: That doesn't make any sense.
**A18.** We changed the sentence as:
*"Beside air-water exchanges, carbon exchanges occur at the water–sediment interface, through biomineralization and/or burial (Regnier et al., 2013b)."[L67-68]*

L106-108: That should go to conclusion and outlook.
**A19.** Indeed, the sentence has been removed from the introduction.

**Materials and methods**

L111: Degree signs needed.
**A20.** We changed decimal coordinates by unit in degree, minute, second.

L114-115: replace "annual water flow" by simply "water flow" because you report any-way the average flow over a longer period, and moreover, you report that flow in volume of water per second, and not per year.
**A21.** Indeed! We changed as recommended

L195-197: Please shortly list the characteristics which are represented.

**A22.** The sentence is now as follows :
*"Here, the Seine drainage network starts from headwater until it fluvial outlet (Poses) and was divided into 69 modeling units, including six axes (axis-object) and 63 upstream basins (basin-object). A map and a table introducing the main characteristics of the modeling units are provided in S2"* [L187-190]

We have also done a new map and a table describing the characteristics of the different modeling units. This description includes: type of modeling units (axis or basin); min and max Strahler orders; drained area; number of river stretches; cumulated length.

L216-219: You state that you could use two variables, DIC and TA, to calculate all other components of the carbonate system. Here I have to disagree. You could be doing this with DIC and carbonate alkalinity, but not with TA which is the sum of carbonate alkalinity and other sources of alkalinity including phosphate, silicate, ammonia and organic ions. But as you represent at least phosphate, ammonia and silicate, you can derive carbonate alkalinity from TA. Is that maybe what the model is doing? If yes, please clarify. But it would mean that you use more variables than just DIC and TA to calculate for instance CO2 concentrations.

**A23.** See our response A3, above in the  #2 Alkalinity section :

In the section 3.4 concerning "pH calculation", we wrongly refer to carbonate alkalinity (CA) instead of Total Alkalinity (TA). This error probably misled the reviewer, making him think that we were recalculating the carbonate alkalinity based on the total alkalinity and ammonium + phosphate ions (which is not the case, we only use TA in our approach, as simplified for freshwater, see above).

L229: "CO2 gradient concentrations" should be "CO2 concentration gradients"
**A24.** We changed the formulation accordingly.

L265-269: How have these studies refined that approach? Did they simply re-calibrate the annual average concentration? Are these average concentrations adapted for different land use types, soil types, etc.? Or do you use only one average concentration per nutrient species which you apply everywhere? Please, clarify.

**A25.** These studies helped refining the approach through new determination of parameters of the kinetics equations, but also using more detailed spatially explicit databases describing for example: lithology, land use, N surplus and the fraction leached according to agricultural statistics.
An average concentration is calculated for each nutrient species at the scale of each modeling unit, taking into account land use, lithology etc. Methodology for calculating these nutrient diffuse sources is specific for each nutrient and described in the literature quoted. We here only detailed the methodology for OC and IC species.

We modify the paragraph to make it clearer:

*"Diffuse sources are calculated at the scale of each modeling units, based on several spatially explicit databases describing natural and anthropogenic constraints on the Seine River basin. Diffuse sources are taken into account by assigning a yearly mean concentration of carbon and nutrients to subsurface and groundwater flow components, respectively."* [L267-270]

L278-281: Are these degradability classes defined somewhere? What is the basic turn-over time or decomposition rate for each class under some sort of standard condition (which needs to be defined)?

**A26.** These degradability classes are described in degradability classes in the book chapter by Billen & Servais (1989). Modélisation des processus de dégradation bactérienne de la matière organique en milieu aquatiques. In : Micro-organismes dans les écosystèmes océaniques (M. Bianchi, Ed), Masson, Paris, page (219-245), and other following papers (e.g. Servais P., Barillier A. & Garnier J. (1995). Determination of the biodegradable fraction of dissolved and particulate organic carbon. Annls Limnol. 31: 75-80).

Here, the fraction of biodegradability were further determined for WWTP effluents, due to the change in treatments, and in new compartments of the hydrosystem (groundwater and small upstream stream).

Reference to Billen & Servais (1989) was added to the text.

For the decomposition rater (turn-over), see our answer just after.

L291-292: Here you should clarify if these degradability classes have the same turnover rates as those for DOC, or if they are defined differently. Otherwise, this statement might be confusing.

**A27.** The fractions of degradability are taken the same for POC and DOC, but the representation of their degradation is different, and parameter of the RIVE model could be found in *(Garnier et al., 2002).*

This precision has been brought:

*"The kinetics for POC and DOC hydrolysis and parameters however are different (Billen and Servais, 1989; Garnier et al., 2002)." [L299-300]*

L293-297: Do you really mean TA here? Or maybe carbonate alkalinity? Note that phosphate, ammonia, silicate and organic ions count into TA.

**A28.** Please refer to our detailed answer about the use of TA in our modeling approach (see **A3**).

L299: You should write mg CO2-C L-1 instead of mg C-CO2 L-1. Figure 3: How can alkalinity be reported in μmol L-1? Do you meanμeq L-1? Also, you should report DIC inμmol L-1 to be consistent, even if you report alkalinity inμeq L-1 (which you definitely should!).

**A29.** As suggested, we modified the 'mg C-$CO_2$-C L$^{-1}$' in 'mg C L$^{-1}$'.

Alkalinity can be report in μmol L$^{-1}$ by dividing the atomic weights of elements by their charges. It is becoming more and more common in to work in μmol L$^{-1}$ (see **A2**).

All our biogeochemical processes are in mgC L$^{-1}$, so we decided to keep $CO_2$ and DIC in mgC L$^{-1}$ to compare them more easily.

L339-340: Could you please give the implied average concentration of free dissolved CO2 for these effluents?

**A30.** The implied average concentration of free dissolved $CO_2$ is 12 mgC L$^{-1}$.

Alshboul et al. (2016) measured CO2 concentrations in WWTP effluents up to 8.5 mgC L$^{-1}$ however these measurements were in German rivers (mean DIC of 20 mgC L$^{-1}$) less carbonated than the Seine River.

L368-370: Did you have additional hydrochemical data available to correct for at least phosphate and ammonia contributions to TA?

**A31.** We do have hydrochemical data for phosphates and ammonia, but according to our use of TA in our modeling approach (see **A3**), we do not use them for correcting TA.

Results

L420: What do you mean by "good levels"?

**A32.** "Good level" means right order of magnitude, which is not trivial, as the model is not calibrated, the value of the parameters being determined independently. However, the wording had been changed as follows:

"*Upstream, within Paris, and downstream of Paris, the model provides simulations in the right order of magnitude of the observed CO$_2$, DIC, TA and pH values, despite the fact that TA was underestimated in the two upstream stations selected for all seasons (Figure 6). DIC and TA simulations followed the observed seasonal patterns with a depletion of concentrations occurring in summer/autumn related to low-flow support by the reservoirs. Indeed, reservoirs showed lower TA and DIC concentrations than rivers (Table 3). In addition to the intra-/inter-stream order variabilities of CO$_2$ (Figure 4), CO$_2$ concentrations showed a wide spread in values over the year (Figure 6). Although simulated CO$_2$ concentrations fitted rather well with the level of the observations (NRMSE = 15%), the model tended to overestimate the winter values upstream and within Paris (Figure 6, left).*" [L428-437]

Figure 6: For the river network within and upstream of Paris, the model shows a very weak performance with regard to seasonality in CO2 concentrations and pH. There appears to be a systematic underestimation of TA throughout time and space. That Would have to be discussed. Moreover, I wonder if a simple recalibration could help to simply solve this problem.

**A33.** The performance of the model has been better described (see the paragraph **A32**).

Recalibration is not the philosophy of the approach. Indeed the principle of the modelling approach is to formalise mathematically the major processes (kinetics equations) from experiments (our own or those from literature) and to determine their parameters independently from the model simulations. Once kinetics and parameters have been a priori fixed on the basis of the current knowledge, the simulations are compared with the observations. A disagreement between simulations and observations may question either the

processes/parameters as represented in the model or/and the quality of the data (in terms of limit conditions and/or observations for validation). Perspectives for improvement are provided in the discussion at several places.

L459-462: Raymond et al. 2012 trained their empirical model for k on relatively small rivers (defined by discharge). As you have discussed before, the equations by Alin et al. may only be valid for a stream width up to 100 m. Also Raymond et al.'s equation is only valid up to a certain discharge. Following that same logic, you cannot apply their equations here. These issues should be discussed here.

**A34.** Indeed, the Raymond et al. equation is not pertinent in high orders; however we decided to keep the formulation for comparison because such $k$ formulation has been widely used in previous research works. Especially, IC budget for the Seine budget provided by Marescaux et al. (2018a) are based on Raymond et al. equation. Keeping a test-simulation (on order 6 and 7) using this equation, allows us to better discuss the differences obtained between this work and previous research work.

But, we totally agree that except for such a comparison, this $k$-value should not be used for high stream orders.

L463-466: As discussed in Alin et al., in wider rivers, wind stress might become the dominant control of k. It seems to be potentially problematic to just omit the term related to wind speed in the equation by Ho et al.. I would expect that the underestimation of k might arise from that. You should quantify that potential bias for a realistic range of wind speed, and discuss why you think that this bias would be negligible. Wouldn't it be better to simply assume an average wind speed? Or you could simply use average monthly wind speed values per stream order from e.g. http://worldclim.org/version2.

**A35.** Indeed, the wind may have a big influence on $k$-value. We only state again that the equation by Ho et al. and O'Connor et al. are only used for SO6 and SO7 and where width > 100m (i.e., less than 400 km of river). Averaging wind by order does not appear relevant here. Also, calculating a mean wind along the main stem of the Seine River seems difficult to use because some sections of the Seine River are highly urbanized and some others are very open. So according to our expertise, implementation of the wind will be considered in our future work, which implies new development in the model. But we thank the reviewer for the database reference that should be useful in the future.

Our previous answer **A5** clearly explains that changing k formulation in these sectors (less than 1.5% of the cumulative length of the Seine network) will lead to a maximum of 5% of change in $CO_2$ emissions from the Seine River.

Consequently, we agree that in these downstream sectors, omitting wind leads to an underestimation of the k, but we also add that this underestimation has very limited impact on our $CO_2$ emissions balance.

L469: Here section 4.1 follows after section 3.1.4.. I have the strong feeling that some sections have gone missing here. But I hope it's just some stupid mistake with numbering. I will simply assume that this is still the results section, and discussion section starts in L550.
A36. Sorry, this is indeed a stupid mistake in numbering. Section 4.1 and 4.2 have become 3.2 and 3.3

L492-493: I assume that "ventilation" means CO2 emissions from water surface. Anyway, you should use a more consistent terminology to not confuse the reader.
A37. Thank you. We changed ventilation by $CO_2$ emissions, in Table 4 included.

Discussion

L554-555: Öquist et al. found that for which river? In how far is that river comparable with the Seine river?

A38. We think that this pattern can be applied to the Seine, because a previous experiment was done for $N_2O$ and showed a similar result (see Garnier et al. 2099, AEE, Fig 5)
We have added this sentence:
"Such a $CO_2$ emission pattern can be applied to the Seine, as a similar result was found for $N_2O$ (Garnier et al., 2009)" [L564-566]

L562-569: You could still calculate the average wind speed per stream order and simply use that in your equation. Also, you could simply adapt k empirically in a way that optimizes the fit between observed and modelled CO2 concentrations.
A39. See the above comment on the wind (A35).

We slightly modify the sentence to clarify that taking wind speed into account in Ho et al. equation could potentially improve the validation of $CO_2$ concentrations (decrease NRMSE) in these downstream sectors (only).

"Regarding gas transfer velocity values, an equation for large rivers with no tidal influence using wind speed could be more appropriate (Alin et al., 2011) and could decrease NRMSE in these downstream sections of the river." [L572-574]

L589-594: Temporal dynamics in CO2 are likely the strongest control on the temporal dynamics of pH. As long as you don't get those right, you won't be able to reproduce pH, no matter what formulation you will use.
A40. We agree with your comments and have deleted the sentence.

L610-618: Here you should mention how much SO1 contributes to the total CO2 emission and to the total stream surface area of the Seine river network. Then you could give the average CO2 emission rate per stream surface area for SO2-SO7 only. Like This, you could support your statement with numbers.

**A41.** Thanks for the suggestion. SO1 represents 9.6% of the Seine River surface area and contributes to 40% of the total $CO_2$ emissions.

[Figure]

We have Add the following sentence:

*"The mapping of $CO_2$ outgassing in the Seine basin clearly showed these spatial trends, with smaller streams releasing more $CO_2$ than median and larger rivers (see Figure 8). Indeed, first-order streams of the Seine River represents 9.6% of the Seine surface area and contributed to 40% of the total $CO_2$ emissions by the river network."* [L651-655]

L684-686: Your results do not support this conclusion. In particular the performance with regard to reproducing observed CO2 concentrations is quite bad, and a decent discussion on why that is missing so far.

**A42.** We understand your remark and we rephrased the conclusions.

However, taking into account that the same biogeochemical model is used from headwaters to the outlet of the river, without tuning the parameters at the scale of the whole basin, it is satisfying to obtain simulation in the correct range of the observed values. We agree that our results call for more work, both in refining the diffuse and point sources, improving the processes taken into account in the model, etc.

---

## Author Comment (AC3) · 11 Feb 2020

**Response to anonymous Referee #3**

**Modeling inorganic carbon dynamics in the Seine River continuum in France by Marescaux et al.**

The authors present a modeling effort of inorganic carbon dynamics in the Seine River. It is done in the pyNuts-Riverstrahler model. With the new module, the outgassing of CO2 is calculated for the time period 2010-2013. Also a budget for inorganic and organic carbon including alkalinity for the whole Seine river basin is presented. The manuscript is well structured. The model performance from small orders to higher orders is reasonable at first sight. However, considering how sensitive the balance between alkalinity – CO2 – pH is, the model performance from small orders to higher orders is impressive. I recommend to publish this paper after major revision.

**Specific comments**

- There are many well tested and well described inorganic carbon modules readily available (see for a review: Orr et al., 2015, https://www.biogeosciences.net/12/1483/2015/bg-12-1483-2015.pdf). Is there a specific reason to develop an own implementation for pyNuts-riverstrahler?

**A0.** This excellent review by Orr et al. 2015 is based on ten packages that aim at calculating ocean carbonate chemistry. The aim of our own implementation was to propose a process-based approach of the modeling $CO_2$ in relationship with the aquatic cycling of nutrients and organic matter, and taking into account the development of micro-organisms (phytoplankton, zooplankton, bacteria) all involved in the aquatic dynamics of $CO_2$ concentrations in the river. A second aim was to route such an inorganic module in an existing drainage network model to calculate $CO_2$ emissions by river.

- In paragraph a kind of sensitivity analyses is presented for the gas transfer velocity. It is not clear to me, why this parameter is chosen. I miss a more extended model sensitivity analyses to determine which input parameters are sensitive to CO2 emissions or carbon export to the sea. Which model parameter contributes most to variability of CO2 emissions?

**A0.** We realized that the tests carried out on the formulation of the k-coefficients (Figure 7), which concern only a very limited downstream part of the Seine system, were not presented in sufficient detail. We have therefore revised the text to better explain these tests on k-formulation, and their impact on total CO2 emissions. see Lines [L463-466], [L580-587] and [L635-639].

We have also carried out additional simulations evaluating the sensitivity of $CO_2$ emissions to different formulations of *k*-values applied to the entire Seine river system. These tests are presented in detail in our A32 answer. In particular, they have allowed a better discussion of the total emissions values obtained, with respect to previous work on the Seine (Marescaux et al., 2018a), and finally strengthen the value of 364 +/- 99 GgC/yr put forward.

**Technical corrections main text**

1. Double equation numbers. Equation numbers in SI and in main article overlap. Please give them different names.

**A1.** Equations in supplementary material are now numbered with S- prefix to prevent overlap.

2. Line 46: The first highlight of a successful implementation. I was surprised by this highlight. There is no word on the implementation details in this article. I think the model itself is never a highlight. The model is a tool to show some of your findings (as you do in this article). So remove.

**A2.** This highlight has been removed and replaced by:

*"$CO_2$ emission from the Seine River was estimated at 364 ± 99 GgC $yr^{-1}$ with the Riverstrahler model." [L46-47]*

3. Line 101: Again purpose of this study is an implementation. I don't think this journal is suited for this purpose.

**A3.** The sentence has been modified as follows:

*"The purpose of the present study was to quantify the sources, transformations, sinks and gaseous emissions of inorganic carbon using the Riverstrahler modelling approach (Billen et al., 1994; Garnier et al., 2002; Thieu et al., 2009)." [L102-104]*

4. Line 102: "pyNuts modeling environment" I would like to have a reference to this. To me it is not clear what the difference is between RiverStrahler, RIVE pyNuts-Riverstrahler. All names are used here. Please elaborate this.

**A4.** We do not refer to "pyNuts modeling environment" in this section anymore. Please refer to our previous answer see A3.

For the differences between RIVE, Riverstrahler and pyNuts modeling environment is now better explained in section 2.2, with adequate references quoted in the text.

For the reviewer information :
- The **RIVE model** aims at representing the biogeochemical functioning of aquatic systems, by simulating concentrations of oxygen, carbon and nutrients (multiple

forms) in relationship with the development of phytoplankton, zooplankton and bacteria. The model also takes into account benthic variables.

- The **Riverstrahler approach** is based a lagrangian description of the circulation of waterbodies within the drainage network. It allows the calculation of geographical and seasonal variations (with a 10 days period resolution) of water flow, water quality and ecological function of a river system based on the biogeochemical RIVE model.
- The **pyNuts modeling environment** is a python framework (with the "Nuts" suffix standing for NUTrientS ) that brings together the biogeochemical modeling code, raw spatially explicit data describing natural and anthropogenic constraints for input calculation, a collection of pre-processing methods and a set of databases structured in a database-management system.

A detailed information could also be found at  https://www.fire.upmc.fr/rive/

5. Line 106: remove s from works

**A5.** The sentence has been removed.

6. Line 111: Add unit to the decimal numbers.

**A6.** The decimal coordinates have been changed into degrees, minutes and seconds.

7. Lines 147 – 154: This footnote is unclear. Last line: calculation of stream velocity. How? Is something fallen of the page here? Use of parameter WSA is confusing. It could mean: mean_width * Slope * Area (not defined here). Change name or put bracket around name.

**A7** Table 1 has been reviewed. It now introduces characteristics of the Seine drainage network until its fluvial outlet (at Poses). The presentation of the formulas for the calculation of mean widths and depths was awkward here, since it has to be done all along the network and not on values averaged by Strahler order. For these two metrics we now use respectively the references Thieu et al. 2009 and Billen et al. 1994.

see the new Table 1 here after :

*Table 1:  Hydro-morphological characteristics of the Seine drainage network, (\*) averaged by Strahler order (SO) and (\*\*) over the time period 2010-2013. Hydrographic network provided by the Agence de l'Eau Seine Normandie and water discharges by the national Banque Hydro database. Depth and flow velocity calculated according to Billen et al 1994; width calculated according to Thieu et al 2009.*

| SO | Draining area km² | Cum. length km | Width (*) m | Depth (**) m | Slope (*) m m⁻¹ | Discharge (**) m3 s⁻¹ | Flow velocity (**) m s⁻¹ |
|----|-------------------|----------------|-------------|--------------|-----------------|----------------------|--------------------------|
| 1 | 36083 | 12759 | 2.4 | 0.14 | 0.01442 | 0.13 | 0.34 |
| 2 | 12354 | 5231 | 5.2 | 0.29 | 0.00540 | 0.66 | 0.36 |
| 3 | 7067 | 2871 | 10.6 | 0.45 | 0.00300 | 2.17 | 0.47 |
| 4 | 4054 | 1548 | 20.2 | 0.79 | 0.00212 | 6.35 | 0.33 |
| 5 | 2649 | 943 | 46.0 | 1.11 | 0.00060 | 25.87 | 0.46 |
| 6 | 2094 | 636 | 77.8 | 2.51 | 0.00029 | 82.22 | 0.42 |
| 7 | 1354 | 318 | 168.3 | 2.61 | 0.00037 | 416.16 | 0.81 |

8. Line 161: Please make figure captions consistent. Figures 1,5,6,.. ends with a dot, but other figure captions not.

**A8.** Figure captions have been homogenized with a systematic dot at the end.

9. Lines 192-197: Message in this paragraph is unclear

**A9.** The Riverstrahler approach applied to any river basin allows to subdivide this basin in sub-basins connected to main axes. Depending on the quality and quantity of available data, the number of simulated objects can vary. For example, the major tributary of the Seine are the upstream Seine Basin, the Marne, and the Oise which could be branched to one axe. But here, because the Seine Basin in well documented, we were able to identify 69 sub-basins, connected to six axes, described per km of stretch.

The paragraph has been re-written as follows:

*"Geomorphology. A drainage network can be described as subbasins (tributaries) connected to one or several main axes, that define a number of modelling units. The modelling approach considers the drainage network as a set of river axes with a spatial resolution of 1 km (axis-object), or they can be aggregated to form subbasins that are idealized as a regular scheme of tributary confluences where each stream order is described by mean characteristics (basin-object). Here, the Seine drainage network starts from headwater until it fluvial outlet (Poses) and was divided into 69 modeling units, including six axes (axis-object) and 63 upstream basins (basin-object). A map and a table introducing the main characteristics of the modeling units are provided in S2." [L182-190]*

10. Line 210: Which module? I only see RIVE in figure 2, including TA and DIC. Highlight the IC module in figure 2.

**A10.** This is exact. We rephased the paragraph as the carbonate system is now fully integrated in the RIVE model.

*"The carbonate system was described by a set of equations (named $CO_2$-module) based on a previous representation provided by Gypens et al. (2004) and adapted for freshwater*

*environments (N. Gypens and A.V. Borges, personal communication). This $CO_2$-module was*
*fully integrated in the RIVE model (Figure 2)." [L212-215]*

11. Line 236: Eq 3 is referred to as eq 1 in SI

**A11.** Yes, the equation of $CO_2$ equilibrium at the air-water is duplicated in the supplementary material in order to facilitate the reading of section S3. We numbered equations in supplementary material with S- prefix to prevent overlap.

12. Line 238: Table 2: It is not clear how column TA is made out of the formulas 3 – 8. Please explain.

**A12.** We calculated for one mole of carbon, how many mole of H+, HCO3- are consumed or produced.

13. Line 258: values and constants are given in Table 2. Is this reference correct? I don't see them.

**A13.** Thank you, we now refer to table S3-1

14. Line 263: Where are the subsurface and groundwater flow components described? Is this in line 201 and further?

**A14.** Exactly, hydrology is described in the paragraph from line 198 to 206. Sub-titles have been added for a better structuration of the section 2.2.

15. Line 296: Are pH values measured? From HCO3- and pH, the CO2 concentrations could be calculated.

**A15.** pH in groundwater is actually measured on a regular basis by French water authorities, but reliability of these measurements seems weak, in particular because we do not have information on pH-meter types used and their calibration, and the way of how piezometers are sampled. . For these reasons, we decided not to use the available pH measurements and to recalculate the pH from DIC and TA concentrations according to Culberson (1980), see S3.4.

16. Line 318: S3. Is this the right reference?

**A16.** Sorry, this is indeed not the right reference. The numbering of the figures and tables was confusing and has been revised.

17. Line 343-351: Reservoirs are an integrated component of the river network itself. They are not point sources, they are receivers of alkalinity. This is a strange paragraph. There is no module with DIC module for reservoirs, so measurements from one reservoir are taken. Does this mean that reservoirs are not part of the module? This can't be true….

**A17.** Thank you for this remark. The title of the section "Point sources from the reservoirs" was poorly chosen. We have changed it by *"Impact of the reservoirs"*.

We indeed have a model of reservoirs using the version of RIVE without the integration of the CO2-module, as we measured only $CO_2$ at 3 occasions in one of the reservoirs. When realizing that the reservoirs have such an impact on the downstream Seine River, not only for nutrients and organic carbon, but also for TA, DIC and hence for $CO_2$ and pH, we used the few reservoir measurements of TA, DIC we measured as forcing variables contrary to the other RIVE variables which were calculated.

One of the sentence of this paragraph has been modified as follows:

*"Owing to the absence of an inorganic carbon module in the modeling of reservoirs yet, we used mean measurements of TA and DIC in reservoirs as forcing variables to the river network."* [L354-356]

18. Line 400: Figure 4: missing x-axes like for example "Strahler order".

**A18.** "Strahler order" has been added as x-axe of figure 4

19. Line 402: Change mgC- CO2 L-1 to mgC L-1

**A19.** Done

20. Line 409: were followed in were followed.

**A20.** Done.

21. Line 438: Figure 6 is too small to see the results.

**A21.** The legends of Figure 6 have been enlarged

22. Line 439: Subscript of CO2 (twice)

**A22.** This is done.

23. Line 440: What is simulation envelope? Can I see this? What is the gray area?

**A23.** Simulation envelope corresponds to standard deviations (gray area). It has been put in evidence in the figure 6 caption.

24. Line 448: Here a time lag is mentioned. But size is total different as well. I don't see any explanation for this.

**A24.** It is true, that phytoplankton dynamics is not well reproduced, although that overall the simulations by the model are in the range of the observations (0.05 to 5 mgC/l, i.e. about 2 to 20 µgChla /l). Tentative explanations are provided in the discussion (line 610-621).

*"Phytoplankton development strongly results from a compromise between dilution rate by the river water and phytoplankton growth rates. But it also depends on nutrients and light availability. Observed water flows are split into two components (runoff and base flow) so*

*that are water flows taken into the model are close to the observations. Nutrients are well reproduced by the model and rarely limiting. However, phytoplankton compartment comprises only 3 groups with their own physiological characteristics (growth rates specifically) and we use empirical relationships for mean daily photosynthetically active radiation (PAR) received per day. Due to major changes in the water quality of the Seine River after the implementation of the Water Framework Directive (from 2000 onwards), phytoplankton biomass was reduced by a factor of 5 to 10, with possibly new groups of phytoplankton not taken into account in the RIVE model. Also instead of using empirical formula for light, observed values of PAR, would certainly improve phytoplankton simulations, especially in February or March when light quickly increase. This could explain the delay in phytoplankton development, which could be probably moved forward while taking into account e.g. a phytoplankton group of small species with a high growth rate (r-strategy), during a short sunny period in winter, often observed".*

25. Line 451: There is a four (number with dot) shown. Delete.

**A25.** Deleted !

26. Line 461: to = too

**A25.** Corrected !

27. Line 522: Subscript of CO2

**A25.** Done !

28. Line 545: Figure 9, to show the spatial dynamics of the ecology in the continuum, it might be interesting to explicitly present the relative contribution of benthic primary producers and the planktonic primary producers to the total primary production.

**A28.** We make the graphic suggested. Indeed, we can observe that benthic respiration is very high in small stream orders and then decrease in medium SOs to re-increase in large SOs. This pattern is described and discussed Lines [543-544]. We did not change the figure in the manuscript because the information is redundant with Figure 8 and we prefer to keep this Figure 9 simple,benthic respiration being included in heterotrophic respiration.

[Figure]

29. Line 563: Did you test the performance of the model with the wind speed parameterization suggested by Alin et al. 2011?

A29. Indeed, the wind may have a big influence on $k$-values. Calculating a mean wind along the main stem of the Seine River seems difficult to use because some sections of the Seine River are highly urbanized and some others are very open. So according to our expertise, implementation of the wind requires new developments that we will investigate in future works.

Nevertheless, we calculated that using different $k$-formulations (namely Raymond, Ho and O'Connor equation) in these sectors (less than 1.5% of the cumulative length of the Seine network) will lead to a 6.2% change in $CO_2$ emissions from the overall Seine River drainage network.

30. Line 584: Any sense of direction which specific algae parameter(s) / trophic condition(s) has/have changed that causes the temporal variability not matching?

A30. Please refer to our answer A24.

31. Line 594: Dot at end of line.

A31. Done

32. Line 602-604: What is the contribution of estimated k-value to the uncertainty of the total basin CO2 emissions? You slightly touch upon in figure 7, but basin total CO2 emissions are not mentioned.

**A32.** New simulations were performed in order to compare the $CO_2$ emission estimates using different k-formulations. In addition to the simulation selected in our manuscript (here call k_Reference), we calculate emissions if the *k* were formulated as:

- Alin et al. (2011) (equation <100 m) (k_Alin) all along the drainage network

- Raymond et al. (2012) (Table 5 Eq. 2) (k_Raymond) all along the drainage network.

The results are presented in the table below. We also add $CO_2$ emissions estimated by Marescaux et al. (2018a) based on observations.

| | Names | k-value SOs 1-6 (width < 100 m) | k-value SOs 6-7 (width > 100 m*) | CO₂ emissions (GgC yr-1) | Time period | Surface area of rivers (km²) |
|---|---|---|---|---|---|---|
| simulations | k_Reference | Alin et al. 2011 (equation < 100 m) | k mix of Ho et al. and O'Connor et al. without the wind term | 364 | 2010-2013 | 260 |
| | k_Alin | Alin et al. 2011 (equation < 100 m) | Alin et al. 2011 (equation < 100 m) | 388 | 2010-2013 | 260 |
| | k_Raymond | Raymond et al. 2012 (Table 2 equation 5) | Raymond et al. 2012 (Table 2 equation 5) | 418 | 2010-2013 | 260 |
| Estimations based on observations | Marescaux_2018 | Raymond et al. 2012 (Table 2 equation 5) | Raymond et al. 2012 (Table 2 equation 5) | 590 | 2010-2016 | 265 |

* SOs 6-7 > 100m represent 367 km out of the 24,306 km of the river network until its outlet at Poses (either 1.5 % only)

**Comparison of the simulations vs. Marescaux et al (2018)**

Our estimates of simulated $CO_2$ outgassing (364 GgC/yr) are lower than our previous estimate based on observation (590 GgC/yr, Marescaux et al. 2018a). This difference is explained below:

- Marescaux et al (2018a) use k- formulation according to Raymond et al. (2012) all along the Seine drainage network (not adapted for large river section) and $CO_2$ emission value is most likely overestimated

- Comparison between the k_Reference, k_Alin and k_Raymond simulations demonstrated that $CO_2$ emissions from the Seine are sensitive to k-formulation (until 15% difference).

- Among the 3 simulations we have compared (k_Reference, k_Alin and k_Raymond), only the k_Reference simulation takes into account a k-formulation adapted for large river sections.

For these reasons, we believe that our estimate of 364 ± 99 GgC/yr, using a process based model, is a more accurate value of $CO_2$ emission from the Seine River. We also acknowledge that this value might be slightly underestimated with respect to Figure 4 (of the present paper)

which shows that our simulated $CO_2$ concentrations were overestimated for SO1 but underestimated for SO2 to SO7.

33. Line 604-606: I would not compare outgassing by surface area to global studies. Reference to temperate rivers are relevant.

**A33.** Thanks, we remove the part on global studies. We rephrased as:

*"The outgassing found for the Seine River by surface area of river of $1400 \pm 381$ gC m$^{-2}$ yr$^{-1}$ is in the middle range of the average estimates of outgassing from temperate rivers (70-2370 gC m$^{-2}$ yr$^{-1}$), including the St. Lawrence River (Yang et al., 1996), Ottawa River (Telmer and Veizer, 1999), Hudson River (Raymond et al., 1997), US temperate rivers (Butman and Raymond, 2011) and Mississippi River (Dubois et al., 2010)."* [L641-645]

34. Lines 613-614: Sentence is not correct.

**A34.** This is right, sorry! We have deleted the incorrect sentence.

35. Line 620-624: The OC export estimate by Meybeck is higher, but the detail and scale of his study is incomparable to yours. How do you know erosion in the Seine is limiting for OC export compared other temperate rivers? Also, what makes the trophic state of the Seine other than other temperate rivers?

**A35** Thanks for your remark, we tried to provide a clearer explanation:

*"Compared with other temperate rivers, the rivers of the northern France, and specifically the Seine River here, are rather flat, their low altitude limiting erosion (Guerrini et al., 1998). In addition, since the implementation of the European Water Framework Directive in the 2000s, decreasing nutrients and carbon in wastewater effluents discharged into the rivers (Rocher and Azimi, 2017), together with a decrease in phytoplankton biomass development (Aissa Grouz et al., 2016; Romero et al., 2016) can explain this difference in DOC fluxes for the Seine, a change probably valid for many other western European rivers (Romero et al., 2013)."* [L658-665]

36. Line 622: change ": )".

**A36.** Done. See new sentence **A35**

37. Line 624: Add ( before Rocher.

**A37.** Done. See new sentence **A35**

38. Line 646: I would add benthic information to figure 9 too

**A38.** Please refer to our previous answer A28

39. Line 660-661: I don't see benthic respiration explicitly mention in figure 9.

**A39.** Indeed, we made a mistake, we refer to figure 8

40. Line 668: Figure8 add blank.

**A40.** Done.

41. Line 693-694: Where do you show small orders are driven by groundwater discharges?

**A41.** We changed the formulation:

*"In small orders, concentrations are mainly driven by diffuse sources."*

This new sentence is clearly supported but Table 4 and figure 8.

**Technical corrections SI**

42. Page 1 and 8 : broken link.

**A42.** Done.

43. Page 2 : *** Now it is added to model RIVE ??

**A43.** Yes these are the new state variable added to the RIVE model when implementing the inorganic carbon module.

44. Figure S1 : I see nine red hatching areas. Not eight. Please change this also in main text (if 9 is the correct number).

**A44.** These numbers have been corrected and we now provided a new map removing part of the Seine River basin flowing downstream it fluvial outlet.

45. Eq 6 does not make sense here. Remove. Will be given in eq 14 and 15.

**A45.** Eq 6 has been removed and the following equations renumbered.

46. Eq 11 : Remove C from K2C

**A46**. This typo has been removed from Eq. 11 (now Eq. S10)

47. Section 3: Eq 17 to 19: What is CA? Carbonic Acid? Carbonate Alkalinity?

**A47.** In section 3, we wrongly refer to carbonate alkalinity (CA) instead of Total Alkalinity (TA). This error has been corrected.

48. Eq. 28 should be $k600 = 13.82 + 0.35v$

**A48.** In the Eq 28 : $k600 = (13.82 + 0.35 * v * 100) / 100$

The multiplication by 100 applies only to v to get water velocity in cm/s. The division by 100 applies to the whole Alin formula to convert the k600 obtained in cm/h into m/h. But thanks to your remark we correct a typo in this section for k600 units (in m/h instead of m/day)

49. Reference list: I would like to have one for the SI and one for the main text. Please also check the reference list. I was looking for Milero et al. 2006. It is used in the text (Table S1), but not mentioned in reference list.

**A49.** This has been done.

---

## Author Comment (AC4) · 14 Feb 2020

You state that you would need only two parameters to implicitly define all elements of the carbonate system, which is basically correct. But you say you would use DIC and TA for that. You could use DIC and carbonate alkalinity to calculate CO2 concentrations, for instance. But using TA instead would lead to erroneous result because of the non-carbonate contributions to TA. I see that you are representing ammonium and phosphate in your model, and it seems like they are included in TA in the model. But It is not clear to me whether you subtract ammonium and phosphate from TA to calculate carbonate alkalinity, and use that to calculate CO2. Here, I would like to see a much more detailed description of how you actually calculate CO2 concentrations and pH, including equations.

Additional answers to the previous answer A3

We completed new simulations and recalculated the CO2 emissions to take into account the remarks on the TA. Indeed, we removed ammonium and phosphate from the total alkalinity when calculating the pH.

| Scenario                                                                 | CO 2 emissions (See Table 4) |
|--------------------------------------------------------------------------|-----------------------------------------|
|                                                                          | $kgC km^{-2} yr^{-1}$                   |
| Reference (see Table 4 of the MS)                                        | 5619                                    |
| Using (CA = Total alkalinity – ammonium – phosphate) to calculate the pH | 5733                                    |

The new simulation showed that taking into account TA or "TA - ammonium - phosphate" to calculate pH (Culberson, 1980) led to a difference in CO2 emissions of less than 2%. This small difference is related to the fact that the Seine basin is a highly carbonated basin where carbonate alkalinity can be approximated by total alkalinity.

---

## Author Response (AR2)

**Response to anonymous Referee #1 (submitted on 08 March 2020)**

**General comments:**

I can see the authors have incorporated the comments from previous review and revised the manuscript accordingly. Improvements can be seen especially in the introduction and discussion.

**A1.** We thank you for having reviewed our manuscript. We are glad that you find the manuscript improved.

However, some fundamental issues have not been solved yet. For example, most of the manuscript focused on the model description and results. But the model is not new as the model has been applied to Seine River network before (see Laruelle et al. 2019), I think the authors need the clarify why you are repeating the work in different years, and what are the new scientific values from this work compared to the previous model applications.

**A2.** The paper of Laruelle et al., 2019 presents some results of a previous simplify version of pyNuts-Riverstrahler model while the earth of their article is focusing on the estuarine modelling (C-GEM). In this new work, we explain in details how we implemented the module by detailing the equations, the inputs and the validation of the river model. Moreover we bring new explanations about the entire Seine river network metabolisms (from Stream Strahler Order 1 to 7) where the other manuscript focuses on a small portion of the Strahler Order 7 and the estuary. As observed and modelled, $CO_2$ concentrations and emissions in/from small streams are not negligible and are essential to understand the Seine River metabolism.

The aims of the two works are thus very different. The methodology completed to describe the inputs (organic or inorganic carbon), the river modelling part and analysis, the metabolism of the Seine River are presented in this present work while the article of Laruelle et al., 2019 is only using the outputs of the pyNuts-Riverstrahler model. In addition, the manuscript of Laruelle et al., 2019 shows simulations only for the year 2010 when in this work, we run and validated the model on the time period 2010-2013.

We selected the time period 2010-2013 to propose a simulation envelope including a dry (2011), a wet (2012) and two years of intermediate hydrological conditions (2010 and 2012). With this time period, we cover the range of hydrological mean conditions that we can observe on the Seine River.

Also, although the author declare they have improved the text writing, there are still issues in the text that lowered the quality of the manuscript. The carbon budget in river systems is important to the global carbon cycle and this paper could make a substantial contribution to this topic, so I encourage the authors to improve the manuscript again to make this work a remarkable one.

**A3.** The manuscript has been already revised by a professional proofreader in order to improve the English writing.

**Specific comments:**

Line 20: process based -> process-based

**A4.** Change has been made

Line 21: supplemented by -> supplemented with

**A5.** Change has been made

Line 35: Metabolism -> Results from metabolism analysis

**A6.** Change has been made

Line 68-71: still hard to understand this statement; eutrophic systems are usually oversaturated with pCO2 respect to atmosphere;

**A7.** We changed the sentence as: [L 68-71] *"As a whole, eutrophic, oligo- and mesotrophic hydrosystems generally act as a source of carbon however, lentic systems may be undersaturated with respect to atmospheric pCO2 (Prairie and Cole, 2009; Xu et al., 2019; Yang et al., 2019)".*

Line 100-101: why not merge this paragraph with previous paragraph? Also, I think the work of Laruelle et al 2019 should be mentioned here as the content of this paper is similar to Laruelle et al. 2019

**A8.** Merge has been made and Laruelle et al. 2019 is now cited as: [L99-L100] *"It is only recently that we investigated pCO2 and emphasized the factors controlling pCO2 dynamics in the Seine River (Marescaux et al., 2018b) or estuary (Laruelle et al., 2019)."*

Line 477-478: these lines are better to go in next section 3.2
**A9.** We let the sentence in the section "3.1" as they concludes the section "3.1." and introduces the next section "3.2.".

Line 725-744: conclusion are a bit rough and just repeating the numbers of results. It is important to highlight the scientific values of your research and why your research is different to others.
**A10.** Thanks for your comment, we removed some results and added a new paragraph:

[L732-736] *"Our Riverstrahler modeling has shown that there are many factors that control $CO_2$ emissions in basins affected by human activity along an aquatic continuum. Once validated by field measurements, which are still too scarce, this generic modeling approach can be applied to any drainage system to better quantify lateral $CO_2$ emission on a continental scale. "*